# Partitioning variability in animal behavioral videos using semi-supervised variational autoencoders

**Matthew R. Whiteway**[1,2,3,4,5]*, **Dan Biderman**[1,2,3,4,5], **Yoni Friedman**[1,6], **Mario Dipoppa**[1,2], **E. Kelly Buchanan**[1,2,3,4,5], **Anqi Wu**[1,2,3,4,5], **John Zhou**[7], **Niccolò Bonacchi**[8], **Nathaniel J. Miska**[9], **Jean-Paul Noel**[10], **Erica Rodriguez**[2,5], **Michael Schartner**[8], **Karolina Socha**[11], **Anne E. Urai**[12], **C. Daniel Salzman**[2,5,13,14,15], **The International Brain Laboratory, John P. Cunningham**[1,2,3,4], **Liam Paninski**[1,2,3,4,5]

**1** Center for Theoretical Neuroscience, Columbia University, New York, New York, United States of America, **2** Mortimer B. Zuckerman Mind Brain Behavior Institute, Columbia University, New York, New York, United States of America, **3** Grossman Center for the Statistics of Mind, Columbia University, New York, New York, United States of America, **4** Department of Statistics, Columbia University, New York, New York, United States of America, **5** Department of Neuroscience, Columbia University, New York, New York, United States of America, **6** Department of Brain and Cognitive Sciences, Massachusetts Institute of Technology, Boston, Massachusetts, United States of America, **7** Department of Computer Science, Columbia University, New York, New York, United States of America, **8** Champalimaud Centre for the Unknown, Lisbon, Portugal, **9** Sainsbury-Wellcome Centre for Neural Circuits and Behavior, University College London, London, United Kingdom, **10** Center for Neural Science, New York University, New York, New York, United States of America, **11** Institute of Ophthalmology, University College London, London, United Kingdom, **12** Cognitive Psychology Unit, Leiden University, Leiden, The Netherlands, **13** Department of Psychiatry, Columbia University, New York, New York, United States of America, **14** New York State Psychiatric Institute, New York, New York, United States of America, **15** Kavli Institute for Brain Sciences, New York, New York, United States of America

* m.whiteway@columbia.edu

**Data Availability Statement:** Code Availability A python/PyTorch implementation of the PS-VAE and MSPS-VAE is available through the Behavenet package, available at https://github.com/

## Abstract

Recent neuroscience studies demonstrate that a deeper understanding of brain function requires a deeper understanding of behavior. Detailed behavioral measurements are now often collected using video cameras, resulting in an increased need for computer vision algorithms that extract useful information from video data. Here we introduce a new video analysis tool that combines the output of supervised pose estimation algorithms (e.g. DeepLabCut) with unsupervised dimensionality reduction methods to produce interpretable, low-dimensional representations of behavioral videos that extract more information than pose estimates alone. We demonstrate this tool by extracting interpretable behavioral features from videos of three different head-fixed mouse preparations, as well as a freely moving mouse in an open field arena, and show how these interpretable features can facilitate downstream behavioral and neural analyses. We also show how the behavioral features produced by our model improve the precision and interpretation of these downstream analyses compared to using the outputs of either fully supervised or fully unsupervised methods alone.

themattinthehatt/behavenet. In addition to the (MS) PS-VAE, the Behavenet package also provides implementations for the VAE and ß-TC-VAE models used in this paper. Please see the Behavenet documentation at https://behavenet.readthedocs.io for more details. A NeuroCAAS (Neuroscience Cloud Analysis As a Service) (Abe et al. 2020) implementation of the PS-VAE can be found at http://www.neurocaas.com/analysis/11. NeuroCAAS replaces the need for expensive computing infrastructure and technical expertise with inexpensive, pay-as-you-go cloud computing and a simple drag-and-drop interface. To fit the PS-VAE, the user simply needs to upload a video, a corresponding labels file, and configuration files specifying desired model parameters. Then, the NeuroCAAS analysis will automatically perform the hyperparameter search as described above, parallelized across multiple GPUs. The output of this process is a downloadable collection of diagnostic plots and videos, as well as the models themselves. See the link provided above for the full details. Data Availability We have publicly released the preprocessed single-session videos, labels, and trained PS-VAE models for this project. The Jupyter notebooks located at https://github.com/themattinthehatt/behavenet/tree/master/examples/ps-vae guide users through downloading the data and models, and performing some of the analyses presented in this paper. head-fixed (IBL) dataset: https://ibl.flatironinstitute.org/public/ps-vae_demo_head-fixed.zip moving mouse dataset: https://figshare.com/articles/dataset/Video_recording_of_a_freely_moving_mouse/16441329/1 mouse face dataset: https://figshare.com/articles/dataset/Video_recording_of_a_mouse_face/13961471/1 two-view dataset: https://figshare.com/articles/dataset/Two_camera_recording_of_a_mouse/14036561/1 The raw data for the head-fixed sessions analyzed with the MSPS-VAE can be accessed through the IBL website. The Jupyter notebook located at https://github.com/themattinthehatt/behavenet/tree/master/examples/msps-vae guides users through downloading and preprocessing the data into the format required by the Behavenet package. Session 1: https://ibl.flatironinstitute.org/public/churchlandlab/Subjects/CSHL047/2020-01-20/001/ Session 2: https://ibl.flatironinstitute.org/public/churchlandlab/Subjects/CSHL049/2020-01-08/001/ Session 3: https://ibl.flatironinstitute.org/public/cortexlab/Subjects/KS023/2019-12-10/001/ Session 4: https://ibl.flatironinstitute.org/public/hoferlab/Subjects/SWC_043/2020-09-21/001/.

**Funding:** This work was supported by the following grants: Gatsby Charitable Foundation GAT3708 (MRW, DB, YF, MD, EKB, AW, JPC, LP; https://

## Author summary

The quantification of animal behavior is a crucial step towards understanding how neural activity produces coordinated movements, and how those movements are affected by genes, drugs, and environmental manipulations. In recent years video cameras have become an inexpensive and ubiquitous way to monitor animal behavior across many species and experimental paradigms. Here we propose a new computer vision algorithm that extracts a succinct summary of an animal's pose on each frame. This summary contains information about a predetermined set of body parts of interest (such as joints on a limb), as well as information about previously unidentified aspects of the animal's pose. Experimenters can thus track body parts they think are relevant to their experiment, and allow the algorithm to discover new dimensions of behavior that might also be important for downstream analyses. We demonstrate this algorithm on videos from four different experimental setups, and show how these new dimensions of behavior can aid in downstream behavioral and neural analyses.

This is a *PLOS Computational Biology* Methods paper.

## Introduction

The ability to produce detailed quantitative descriptions of animal behavior is driving advances across a wide range of research disciplines, from genetics and neuroscience to psychology and ecology [1–6]. Traditional approaches to quantifying animal behavior rely on time consuming and error-prone human video annotation, or constraining the animal to perform simple, easy to measure actions (such as reaching towards a target). These approaches limit the scale and complexity of behavioral datasets, and thus the scope of their insights into natural phenomena [7]. These limitations have motivated the development of new high-throughput methods which quantify behavior from videos, relying on recent advances in computer hardware and computer vision algorithms [8, 9].

The automatic estimation of animal posture (or "pose") from video data is a crucial first step towards automatically quantifying behavior in more naturalistic settings [10–13]. Modern pose estimation algorithms rely on supervised learning: they require the researcher to label a relatively small number of frames (tens to hundreds, which we call "human labels"), indicating the location of a predetermined set of body parts of interest (e.g. joints). The algorithm then learns to label the remaining frames in the video, and these pose estimates (which we refer to simply as "labels") can be used for downstream analyses such as quantifying behavioral dynamics [13–17] and decoding behavior from neural activity [18, 19]. One advantage of these supervised methods is that they produce an inherently interpretable output: the location of the labeled body parts on each frame. However, specifying a small number of body parts for labeling will potentially miss some of the rich behavioral information present in the video, especially if there are features of the pose important for understanding behavior that are not known *a priori* to the researcher, and therefore not labeled. Furthermore, it may be difficult to accurately label and track body parts that are often occluded, or are not localizable to a single point in space, such as the overall pose of the face, body, or hand.

A complementary approach for analyzing behavioral videos is the use of fully unsupervised dimensionality reduction methods. These methods do not require human labels (hence, unsupervised), and instead model variability across all pixels in a high-dimensional behavioral

www.gatsby.org.uk/), McKnight Foundation (JPC; https://www.mcknight.org/), Helen Hay Whitney Fellowship (ER; http://hhwf.org/research-fellowship), German National Academy of Sciences Leopoldina (AEU; https://www.leopoldina.org/), International Brain Research Organization (AEU; https://ibro.org/), NSF DBI-1707398 (MRW, DB, YF, MD, EKB, AW, JPC, LP; https://nsf.gov/), NIH R21MH116348 (CDS; https://www.nih.gov/), NIH RF1MH120680 (LP; https://www.nih.gov/), NIH T32NS064929 (EKB; https://www.nih.gov/), NIH T32MH015144 (ER; https://www.nih.gov/), NIH U19NS107613 (MRW, YF, MD, EKB, AW, LP; https://www.nih.gov/), NIH UF1NS107696 (LP; https://www.nih.gov/), Simons Foundation 542963 (DB, AW, JPC; https://www.simonsfoundation.org/), Simons Foundation 543023 (MRW, AW, NJM, JPN, MS, KS, LP; https://www.simonsfoundation.org/), Wellcome Trust 209558 (NB, NJM, MS, LP; https://wellcome.org/), and Wellcome Trust 216324 (NB, NJM, MS, LP; https://wellcome.org/). The funders had no role in study design, data collection and analysis, decision to publish, or preparation of the manuscript.

**Competing interests:** The authors have declared that no competing interests exist.

video with a small number of hidden, or "latent" variables; we refer to the collection of these latent variables as the "latent representation" of behavior. Linear unsupervised dimensionality reduction methods such as Principal Component Analysis (PCA) have been successfully employed with both video [20–23] and depth imaging data [24, 25]. More recent work performs video compression using nonlinear autoencoder neural networks [26, 27]; these models consist of an "encoder" network that compresses an image into a latent representation, and a "decoder" network which transforms the latent representation back into an image. Especially promising are convolutional autoencoders, which are tailored for image data and hence can extract a compact latent representation with minimal loss of information. The benefit of this unsupervised approach is that, by definition, it does not require human labels, and can therefore capture a wider range of behavioral features in an unbiased manner. The drawback to the unsupervised approach, however, is that the resulting low-dimensional latent representation is often difficult to interpret, which limits the specificity of downstream analyses.

In this work we seek to combine the strengths of these two approaches by finding a low-dimensional, latent representation of animal behavior that is partitioned into two subspaces: a supervised subspace, or set of dimensions, that is required to directly reconstruct the labels obtained from pose estimation; and an orthogonal unsupervised subspace that captures additional variability in the video not accounted for by the labels. The resulting semi-supervised approach provides a richer and more interpretable representation of behavior than either approach alone.

Our proposed method, the Partitioned Subspace Variational Autoencoder (PS-VAE), is a semi-supervised model based on the fully unsupervised Variational Autoencoder (VAE) [28, 29]. The VAE is a nonlinear autoencoder whose latent representations are probabilistic. Here, we extend the standard VAE model in two ways. First, we explicitly require the latent representation to contain information about the labels through the addition of a discriminative network that decodes the labels from the latent representation [30–37]. Second, we incorporate an additional term in the PS-VAE objective function that encourages each dimension of the unsupervised subspace to be statistically independent, which can provide a more interpretable latent representation [38–44]. There has been considerable work in the VAE literature for endowing the latent representation with semantic meaning. Our PS-VAE model is distinct from all existing approaches but has explicit mathematical connections to these, especially to [37, 45]. We provide a high-level overview of the PS-VAE in the following section, and in-depth mathematical exposition in the Methods. We then contextualize our work within related machine learning approaches in S1 Appendix.

We first apply the PS-VAE to a head-fixed mouse behavioral video [46]. We track paw positions and recover unsupervised dimensions that correspond to jaw position and local paw configuration. We then apply the PS-VAE to a video of a mouse freely moving around an open field arena. We track the ears, nose, back, and tail base, and recover unsupervised dimensions that correspond to more precise information about the pose of the body. We then demonstrate how the PS-VAE enables downstream analyses on two additional head-fixed mouse neuro-behavioral datasets. The first is a close up video of a mouse face (a similar setup to [47]), where we track pupil area and position, and recover unsupervised dimensions that separately encode information about the eyelid and the whisker pad. We then use this interpretable behavioral representation to construct separate saccade and whisking detectors. We also decode this behavioral representation with neural activity recorded from visual cortex using two-photon calcium imaging, and find that eye and whisker information are differentially decoded. The second dataset is a two camera video of a head-fixed mouse [22], where we track moving mechanical equipment and one visible paw. The PS-VAE recovers unsupervised dimensions that correspond to chest and jaw positions. We use this interpretable behavioral representation

to separate animal and equipment movement, construct individual movement detectors for the paw and body, and decode the behavioral representation with neural activity recorded across dorsal cortex using widefield calcium imaging. Importantly, we also show how the uninterpretable latent representations provided by a standard VAE do not allow for the specificity of these analyses in both example datasets. These results demonstrate how the interpretable behavioral representations learned by the PS-VAE can enable targeted downstream behavioral and neural analyses using a single unified framework. Finally, we extend the PS-VAE framework to accommodate multiple videos from the same experimental setup by introducing a new subspace that captures variability in static background features across videos, while leaving the original subspaces (supervised and unsupervised) to capture dynamic behavioral features. We demonstrate this extension on multiple videos from the head-fixed mouse experimental setup [46]. A python/PyTorch implementation of the PS-VAE is available on github as well as the NeuroCAAS cloud analysis platform [48], and we have made all datasets publicly available; see the Data Availability and Code Availability statements for more details.

## PS-VAE model formulation

The goal of the PS-VAE is to find an interpretable, low-dimensional latent representation of a behavioral video. Both the interpretability and low dimensionality of this representation make it useful for downstream modeling tasks such as learning the dynamics of behavior and connecting behavior to neural activity, as we show in subsequent sections. The PS-VAE makes this behavioral representation interpretable by partitioning it into two sets of latent variables: a set of supervised latents, and a separate set of unsupervised latents. The role of the supervised latents is to capture specific features of the video that users have previously labeled with pose estimation software, for example joint positions. To achieve this, we require the supervised latents to directly reconstruct a set of user-supplied labels. The role of the unsupervised subspace is to then capture behavioral features in the video that have not been previously labeled. To achieve this, we require the full set of supervised and unsupervised latents to reconstruct the original video frames. We briefly outline the mathematical formulation of the PS-VAE here; full details can be found in the Methods, and we draw connections to related work from the machine learning literature in S1 Appendix.

The PS-VAE is an autoencoder neural network model that first compresses a video frame $\mathbf{x}$ into a low-dimensional vector $\boldsymbol{\mu}(\mathbf{x}) = f(\mathbf{x})$ through the use of a convolutional encoder neural network $f(\cdot)$ (Fig 1). We then proceed to partition $\boldsymbol{\mu}(\mathbf{x})$ into supervised and unsupervised subspaces, respectively defined by the linear transformations $A$ and $B$. We define the supervised representation as

$$\mathbf{z}_s = A\boldsymbol{\mu}(\mathbf{x}) + \boldsymbol{\epsilon}_{z_s},  \tag{1}$$

where $\boldsymbol{\epsilon}_{z_s}$ (and subsequent $\boldsymbol{\epsilon}$ terms) denotes Gaussian noise, which captures the fact that $A\boldsymbol{\mu}(\mathbf{x})$ is merely an estimate of $\mathbf{z}_s$ from the observed data. We refer to $\mathbf{z}_s$ interchangeably as the "supervised representation" or the "supervised latents." We construct $\mathbf{z}_s$ to have the same number of elements as there are label coordinates $\mathbf{y}$, and enforce a one-to-one element-wise linear mapping between the two, as follows:

$$\mathbf{y} = D\mathbf{z}_s + \mathbf{d} + \boldsymbol{\epsilon}_y,  \tag{2}$$

where $D$ is a diagonal matrix that scales the coordinates of $\mathbf{z}_s$ without mixing them, and $\mathbf{d}$ is an offset term. [Note we could easily absorb the diagonal matrix in to the linear mapping $A$ from Eq 1, but we instead separate these two so that we can treat the random variable $\mathbf{z}_s$ as a latent variable with a known prior such as $\mathcal{N}(0, 1)$ which does not rely on the magnitude of the label

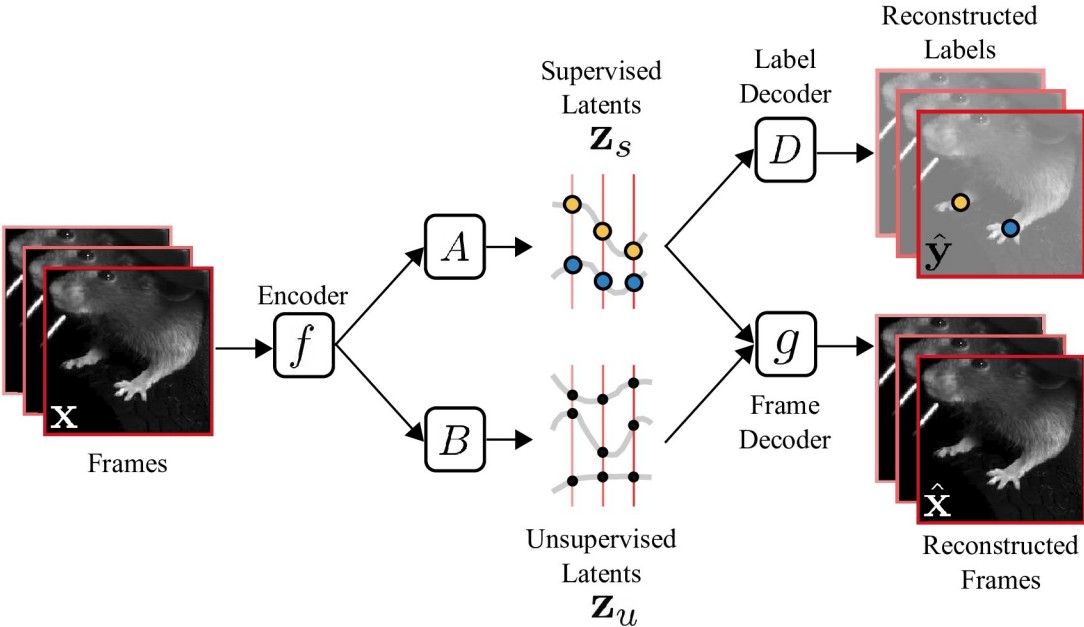

**Fig 1. Overview of the Partitioned Subspace VAE (PS-VAE).** The PS-VAE takes a behavioral video as input and finds a low-dimensional latent representation that is partitioned into two subspaces: one subspace contains the *supervised* latent variables $\mathbf{z}_s$, and the second subspace contains the *unsupervised* latent variables $\mathbf{z}_u$. The supervised latent variables are required to reconstruct user-supplied labels, for example from pose estimation software (e.g. DeepLabCut [10]). The unsupervised latent variables are then free to capture remaining variability in the video that is not accounted for by the labels. This is achieved by requiring the combined supervised and unsupervised latents to reconstruct the video frames. An additional term in the PS-VAE objective function factorizes the distribution over the unsupervised latents, which has been shown to result in more interpretable latent representations [45].

values.] Thus, Eq 2 amounts to a multiple linear regression predicting $\mathbf{y}$ from $\mathbf{z}_s$ with no interaction terms.

Next we define the unsupervised representation as

$$\mathbf{z}_u = B\boldsymbol{\mu}(\mathbf{x}) + \boldsymbol{\epsilon}_{z_u}, \tag{3}$$

recalling that $B$ defines the unsupervised subspace. We refer to $\mathbf{z}_u$ interchangeably as the "unsupervised representation" or the "unsupervised latents."

We now construct the full latent representation $\mathbf{z} = [\mathbf{z}_s; \mathbf{z}_u]$ through concatenation and use $\mathbf{z}$ to reconstruct the observed video frame through the use of a convolutional decoder neural network $g(\cdot)$:

$$\mathbf{x} = g(\mathbf{z}) + \boldsymbol{\epsilon}_x. \tag{4}$$

We take two measures to further encourage interpretability in the unsupervised representation $\mathbf{z}_u$. The first measure ensures that $\mathbf{z}_u$ does not contain information from the supervised representation $\mathbf{z}_s$. One approach is to encourage the mappings $A$ and $B$ to be orthogonal to each other. In fact we go one step further and encourage the entire latent space to be orthogonal by defining $U = [A; B]$ and adding the penalty term $\|UU^T - I\|$ to the PS-VAE objective function (where $I$ is the identity matrix). This orthogonalization of the latent space is similar to PCA, except we do not require the dimensions to be ordered by variance explained. However, we do retain the benefits of an orthogonalized latent space, which will allow us to modify one latent coordinate without modifying the remaining coordinates, facilitating interpretability [37].

The second measure we take to encourage interpretability in the unsupervised representation is to maximize the statistical independence between the dimensions. This additional measure is necessary because even when we represent the latent dimensions with a set of orthogonal vectors, the distribution of the latent variables within this space can still contain correlations (e.g. Fig 2B, *top*). To minimize correlation, we take an information-theoretic approach and penalize for the "Total Correlation" metric as proposed by [42] and [45]. Total Correlation is a generalization of mutual information to more than two random variables, and is defined as the Kullback-Leibler (KL) divergence between a joint distribution $p(z_1, \ldots, z_D)$ and a factorized version of this distribution $p(z_1) \ldots p(z_D)$. Our penalty encourages the joint multivariate latent distribution to be factorized into a set of independent univariate distributions (e.g. Fig 2B, *bottom*).

The final PS-VAE objective function contains terms for label reconstruction, frame reconstruction, orthogonalization of the full latent space, and the statistical independence between $\mathbf{z}_u$'s factors. The model requires several user-provided hyperparameters, and in the Methods we provide guidance on how to set these. One important hyperparameter is the dimensionality of the unsupervised subspace. In the following sections we use 2D unsupervised subspaces, because these are easy to visualize and the resulting models perform well empirically. At several points we explore models with larger subspaces. In general we recommend starting with a 2D subspace, then increasing one dimension at a time until the results are satisfactory. We emphasize that there is no single correct value for this hyperparameter; what constitutes a satisfactory result will depend on the data and the desired downstream analyses.

## Results

### Application of the PS-VAE to a head-fixed mouse dataset

We first apply the PS-VAE to an example dataset from the International Brain Lab (IBL) [46], where a head-fixed mouse performs a visual decision-making task by manipulating a wheel with its fore paws. We tracked the left and right paw locations using Deep Graph Pose [13]. First, we quantitatively demonstrate the model successfully learns to reconstruct the labels, and then we qualitatively demonstrate the model's ability to learn interpretable representations by exploring the correspondence between the extracted latent variables and reconstructed frames. For the results shown here, we used models with a 6D latent space: a 4D supervised subspace (two paws, each with $x$ and $y$ coordinates) and a 2D unsupervised subspace. S1 Table details the complete hyperparameter settings for each model, and in the Methods we explore the selection and sensitivity of these hyperparameters.

**Model fits.** We first investigate the supervised representation of the PS-VAE, which serves two useful purposes. First, by forcing this representation to reconstruct the labels, we ensure these dimensions are interpretable. Second, we ensure the latent representation contains information about these known features in the data, which may be overlooked by a fully unsupervised method. For example, the pixel-wise mean square error (MSE) term in the standard VAE objective function will only allow the model to capture features that drive a large amount of pixel variance. However, meaningful features of interest in video data, such as a pupil or individual fingers on a hand, may only drive a small amount of pixel variance. As long as these features themselves change over time, we can ensure they are represented in the latent space of the model by tracking them and including them in the supervised representation. Rare behaviors (such as a single saccade) will be more difficult for the model to capture.

We find accurate label reconstruction (Fig 2C, blue lines), with $R^2 = 0.85 \pm 0.01$ (mean ± s.e.m) across all held-out test data. This is in contrast to a standard VAE, whose latent variables are much less predictive of the labels; to show this, we first fit a standard VAE model with 6

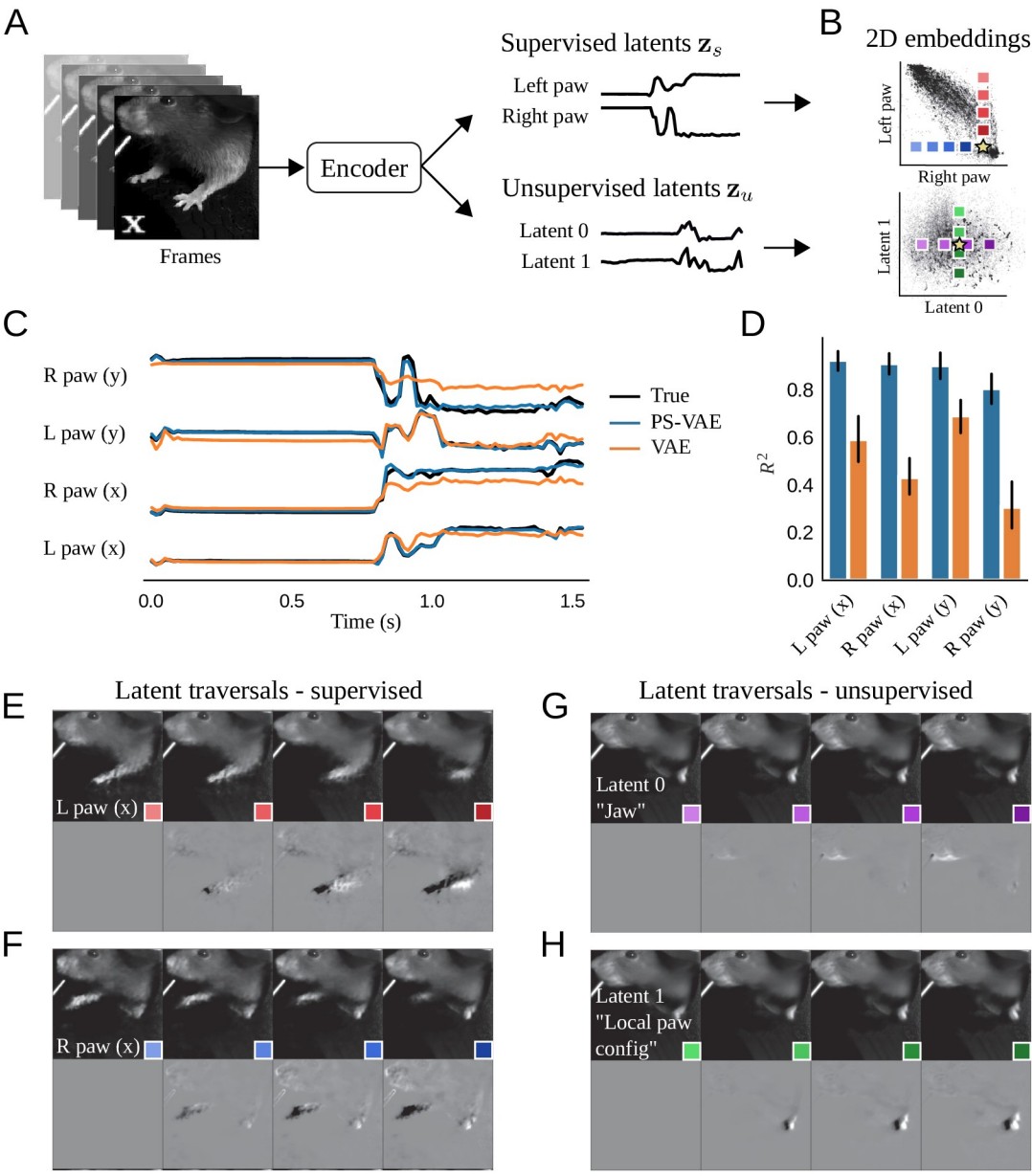

**Fig 2. The PS-VAE successfully partitions the latent representation of a head-fixed mouse video [46].** The dataset contains labels for each fore paw. **A**: The PS-VAE transforms frames from the video into a set of supervised latents $\mathbf{z}_s$ and unsupervised latents $\mathbf{z}_u$. **B**: *Top*: A visualization of the 2D embedding of supervised latents corresponding to the horizontal coordinates of the left and right paws. *Bottom*: The 2D embedding of the unsupervised latents. **C**: The true labels (black lines) are almost perfectly reconstructed by the supervised subspace of the PS-VAE (blue lines). We also reconstruct the labels from the latent representation of a standard VAE (orange lines), which captures some features of the labels but misses much of the variability. **D**: Observations from the trial in C hold across all labels and test trials. Error bars represent a 95% bootstrapped confidence interval over test trials. **E**: To investigate individual dimensions of the latent representation, frames are generated by selecting a test frame (yellow star in B), manipulating the latent representation one dimension at a time, and pushing the resulting representation through the frame decoder. *Top*: Manipulation of the $x$ coordinate of the left paw. Colored boxes indicate the location of the corresponding point in the latent space from the top plot in B. Movement along this (red) dimension results in horizontal movements of the left paw. *Bottom*: To better visualize subtle differences between the frames above, the left-most frame is chosen as a base frame from which all frames are subtracted. **F**: Same as E except the manipulation is performed with the $x$ coordinate of the right paw. **G, H**: Same as E, F except the manipulation is performed in the two unsupervised dimensions. Latent 0 encodes the position of the jaw line, while Latent 1 encodes the local configuration (rather than absolute position) of the left paw. See S6 Video for a dynamic version of these traversals. See S1 Table for information on the hyperparameters used in the models for this and all subsequent figures.

latents, then fit a post-hoc linear regression model from the latent space to the labels (Fig 2C, orange lines). While this regression model is able to capture substantial variability in the labels ($R^2 = 0.55 \pm 0.02$), it still fails to perform as well as the PS-VAE (Fig 2D). We also fit a post-hoc *non*linear regression model in the form of a multi-layer perceptron (MLP) neural network, which performed considerably better ($R^2 = 0.83 \pm 0.01$). This performance shows that the VAE latents do in fact contain significant information about the labels, but much of this information is not linearly decodable. This makes the representation more difficult to use for some downstream analyses, which we address below. The supervised PS-VAE latents, on the other hand, are linearly decodable by construction.

Next we investigate the degree to which the PS-VAE partitions the supervised and unsupervised subspaces. Ideally the information contained in the supervised subspace (the labels) will not be represented in the unsupervised subspace. To test this, we fit a post-hoc linear regression model from the *unsupervised* latents $\mathbf{z}_u$ to the labels. This regression has poor predictive power ($R^2 = 0.07 \pm 0.03$), so we conclude that there is little label-related information contained in the unsupervised subspace, as desired.

We now turn to a qualitative assessment of how well the PS-VAE produces interpretable representations of the behavioral video. In this context, we define an "interpretable" (or "disentangled") representation as one in which each dimension of the representation corresponds to a single factor of variation in the data, e.g. the movement of an arm, or the opening/closing of the jaw. To demonstrate the PS-VAE's capacity to learn interpretable representations, we generate novel video frames from the model by changing the latent representation one dimension at a time—which we call a "latent traversal"—and visually compare the outputs [37–39, 42–44]. If the representation is sufficiently interpretable (and the decoder has learned to use this representation), we should be able to easily assign semantic meaning to each latent dimension.

The latent traversal begins by choosing a test frame and pushing it through the encoder to produce a latent representation (Fig 2A). We visualize the latent representation by plotting it in both the supervised and unsupervised subspaces, along with all the training frames (Fig 2B, *top* and *bottom*, respectively; the yellow star indicates the test frame, black points indicate all training frames). Next we choose a single dimension of the representation to manipulate, while keeping the value of all other dimensions fixed. We set a new value for the chosen dimension, say the 20th percentile of the training data. We can then push this new latent representation through the frame decoder to produce a generated frame that should look like the original, except for the behavioral feature represented by the chosen dimension. Next we return to the latent space and pick a new value for the chosen dimension, say the 40th percentile of the training data, push this new representation through the frame decoder, and repeat, traversing the chosen dimension. Traversals of different dimensions are indicated by the colored boxes in Fig 2B. If we look at all of the generated frames from a single traversal next to each other, we expect to see smooth changes in a single behavioral feature.

We first consider latent traversals of the supervised subspace. The *y*-axis in Fig 2B (*top*) putatively encodes the horizontal position of the left paw; by manipulating this value—and keeping all other dimensions fixed—we expect to see the left paw move horizontally in the generated frames, while all other features (e.g. right paw) remain fixed. Indeed, this latent space traversal results in realistic looking frames with clear horizontal movements of the left paw (Fig 2E, *top*). The colored boxes indicate the location of the corresponding latent representation in Fig 2B. As an additional visual aid, we fix the left-most generated frame as a base frame and replace each frame with its difference from the base frame (Fig 2E, *bottom*). We find similar results when traversing the dimension that putatively encodes the horizontal position of the

right paw (Fig 2F), thus demonstrating the supervised subspace has adequately learned to encode the provided labels.

The representation in the unsupervised subspace is more difficult to validate since we have no *a priori* expectations for what features the unsupervised representation should encode. We first note that the facial features are better reconstructed by the full PS-VAE than when using the labels alone, hinting at the features captured by the unsupervised latents (see S1 Video for the frame reconstruction video, and S14 Fig for panel captions). We repeat the latent traversal exercise once more by manipulating the representation in the unsupervised subspace. Traversing the horizontal (purple) dimension produces frames that at first appear all the same (Fig 2G, *top*), but when looking at the differences it becomes clear that this dimension encodes jaw position (Fig 2G, *bottom*). Similarly, traversal of the vertical (green) dimension reveals changes in the local configuration of the left paw (Fig 2H). It is also important to note that none of these generated frames show large movements of the left or right paws, which should be fully represented by the supervised subspace. See S6 Video for a dynamic version of these traversals, and S15 Fig for panel captions. The PS-VAE is therefore able to find an interpretable unsupervised representation that does not qualitatively contain information about the supervised representation, as desired.

**Qualitative model comparisons.** We now utilize the latent traversals to highlight the role that the label reconstruction term and the Total Correlation (TC) term of the PS-VAE play in producing an interpretable latent representation. We first investigate a standard VAE as a baseline model, which neither reconstructs the labels nor penalizes the TC among the latents. We find that many dimensions in the VAE representation simultaneously encode both the paws and the jaw (S10 Video). Next we fit a model that contains the TC term, but does not attempt to reconstruct the labels. This model is the $\beta$-TC-VAE of [45]. As in the PS-VAE, the TC term orients the latent space by producing a factorized representation, with the goal of making it more interpretable. [Note, however, the objective still acts to explain pixel-level variance, and thus will not find features that drive little pixel variance.] The $\beta$-TC-VAE traversals (S11 Video) reveal dimensions that mostly encode movement from one paw or the other, although these do not directly map on the $x$ and $y$ coordinates as the supervised subspace of the PS-VAE does. We also find a final dimension that encodes jaw position. The TC term therefore acts to shape the unsupervised subspace in a manner that may be more interpretable to human observers. Finally, we consider a PS-VAE model which incorporates label information but does not penalize the TC term. The traversals (S7 Video) reveal each unsupervised latent contains (often similar) information about the paws and jaw, and thus the model has lost interpretability in the unsupervised latent space. However, the supervised latents still clearly represent the desired label information, and thus the model has maintained interpretability in the supervised latent space; indeed, the label reconstruction ability of the PS-VAE is not affected by the TC term across a range of hyperparameter settings (Methods). These qualitative evaluations demonstrate that the label reconstruction term influences the interpretability of the supervised latents, while the TC term influences the interpretability of the unsupervised latents.

We now briefly explore PS-VAE models with more than two unsupervised dimensions, and keep all other hyperparameters the same as the best performing 2D model. We first move to a 3D unsupervised subspace, and find two dimensions that correspond to those we found in the 2D model: a jaw dimension, and a left paw configuration dimension. The third dimension appears to encode the position of the elbows: traversing this dimension shows the elbows moving inwards and outwards together (S8 Video). We then move to a 4D unsupervised subspace, and find three dimensions that correspond to those in the 3D model, and a fourth dimension that is almost an exact replica of the elbow dimension (S9 Video). Additional models fit with

different weight initializations produced the same behavior: three unique dimensions matching those of the 3D model, and a fourth dimension that matched one of the previous three. It is possible that adjusting the hyperparameters could lead to four distinct dimensions (for example by increasing the weight on the TC term), but we did not explore this further. We conclude that the PS-VAE can reliably find a small number of interpretable unsupervised dimensions, but the possibility of finding a larger number of unique dimensions will depend on the data and other hyperparameter settings.

### Application of the PS-VAE to a freely moving mouse dataset

We next apply the PS-VAE to a video of a mouse in an open field arena—another ubiquitous behavioral paradigm [49]—to demonstrate how our model generalizes beyond the head-fixed preparation. We tracked the nose, ears, back, and tail base using DeepLabCut [10], and cropped and rotated the mouse to obtain egocentric alignment across frames (Fig 3A). For our analysis we use a PS-VAE model with a 10D latent space: an 8D supervised subspace ($x$, $y$ coordinates for five body parts, minus two $y$ coordinates that are fixed by our alignment procedure) and a 2D unsupervised subspace.

The PS-VAE is able to reconstruct the labels reasonably well ($R^2 = 0.68 \pm 0.02$), slightly outperforming the linear and nonlinear regressions from the VAE latents ($R^2 = 0.59 \pm 0.03$ and $R^2 = 0.65 \pm 0.02$, respectively) (Fig 3B and 3C). This dataset also highlights how the PS-VAE can handle missing label data. Tracking may occasionally be lost, for example when a body part is occluded, and pose estimation packages like DLC and DGP will typically assign low likelihood values to the corresponding labels on such frames. During the PS-VAE training, all labels that

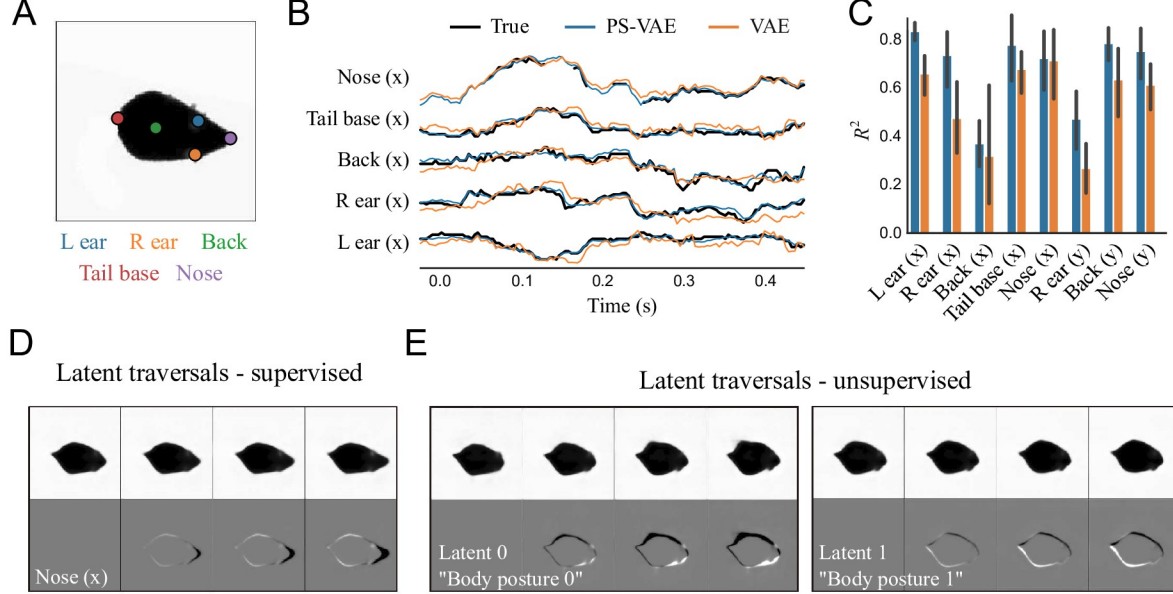

**Fig 3. The PS-VAE successfully partitions the latent representation of a freely moving mouse video. A**: Example frame from the video. The indicated points are tracked to provide labels for the PS-VAE supervised subspace. **B**: The true labels (black lines) and their reconstructions from the PS-VAE supervised subspace (blue lines) and a standard VAE (orange lines); both models are able to capture much of the label variability. The PS-VAE is capable of interpolating missing labels, as seen in the "Nose (x)" trace; see text for details. Only $x$ coordinates are shown to reduce clutter. **C**: Observations from the example trial hold across all labels and test trials. The $y$ coordinates of the left ear and tail base are missing because these labels are fixed by our egocentric alignment procedure. Error bars are computed as in Fig 2D. **D**: Frames generated by manipulating the latent corresponding to the $x$ coordinate of the nose in the supervised subspace. **E**: Same as panel D except the manipulation is performed in the two unsupervised dimensions. These latents capture more detailed information about the body posture than can be reconstructed from the labels. See S12 Video for a dynamic version of these traversals.

fall below an arbitrary likelihood threshold of 0.9 are simply omitted from the computation of the label reconstruction term and its gradient (likelihood values lie in the range [0, 1]). Nevertheless, the supervised subspace of the PS-VAE will still produce an estimate of the label values for these omitted time points, which allows the model to interpolate through missing label information (Fig 3B, "Nose (x)" trace).

In addition to the good label reconstructions, the latent traversals in the supervised subspace also show the PS-VAE learned to capture the label information (Fig 3D). While the supervised latents capture much of the pose information, we find there are additional details about the precise pose of the body that the unsupervised latents capture (see S2 Video for the frame reconstruction video). The latent traversals in the unsupervised subspace show these dimensions have captured complementary (Fig 3E) and uncorrelated (S10H Fig) information about the body posture (S12 Video). To determine whether or not these unsupervised latents also encode label-related information, we again fit a post-hoc linear regression model from the unsupervised latents to the labels. This regression has poor predictive power ($R^2 = 0.08 \pm 0.03$), which verifies the unsupervised latent space is indeed capturing behavioral variability not accounted for by the labels.

## The PS-VAE enables targeted downstream analyses

The previous section demonstrated how the PS-VAE can successfully partition variability in behavioral videos into a supervised subspace and an interpretable unsupervised subspace. In this section we turn to several downstream applications using different datasets to demonstrate how this partitioned subspace can be exploited for behavioral and neural analyses. For each dataset, we first characterize the latent representation by showing label reconstructions and latent traversals. We then quantify the dynamics of different behavioral features by fitting movement detectors to selected dimensions in the behavioral representation. Finally, we decode the individual behavioral features from simultaneously recorded neural activity. We also show how these analyses are not possible with the "entangled" representations produced by the VAE.

**A close up mouse face video.** The first example dataset is a close up video of a mouse face (Fig 4A), recorded while the mouse quietly sits and passively views drifting grating stimuli (setup is similar to [47]). We tracked the pupil location and pupil area using Facemap [50]. For our analysis we use models with a 5D latent space: a 3D supervised subspace ($x$, $y$ coordinates of pupil location, and pupil area) and a 2D unsupervised subspace.

The PS-VAE is able to successfully reconstruct the pupil labels ($R^2 = 0.71 \pm 0.02$), again outperforming the linear regression from the VAE latents ($R^2 = 0.27 \pm 0.03$) (Fig 4B and 4C). The difference in reconstruction quality is even more pronounced here than the previous datasets because the feature that we are tracking—the pupil—is composed of a small number of pixels, and thus is not (linearly) captured well by the VAE latents. Furthermore, in this dataset we do not find a substantial improvement when using nonlinear MLP regression from the VAE latents ($R^2 = 0.31 \pm 0.01$), indicating that the VAE ignores much of the pupil information altogether. The latent traversals in the supervised subspace show the PS-VAE learned to capture the pupil location, although correlated movements at the edge of the eye are also present, especially in the horizontal ($x$) position (Fig 4D; pupil movements are more clearly seen in the traversal video S14 Video). The latent traversals in the unsupervised subspace show a clear separation of the whisker pad and the eyelid (Fig 4E). To further validate these unsupervised latents, we compute a corresponding 1D signal for each latent by cropping the frames around the whisker pad or the eyelid, and taking the first PCA component as a "hand engineered" representation of that behavioral feature. The PS-VAE latents show a reasonable correlation

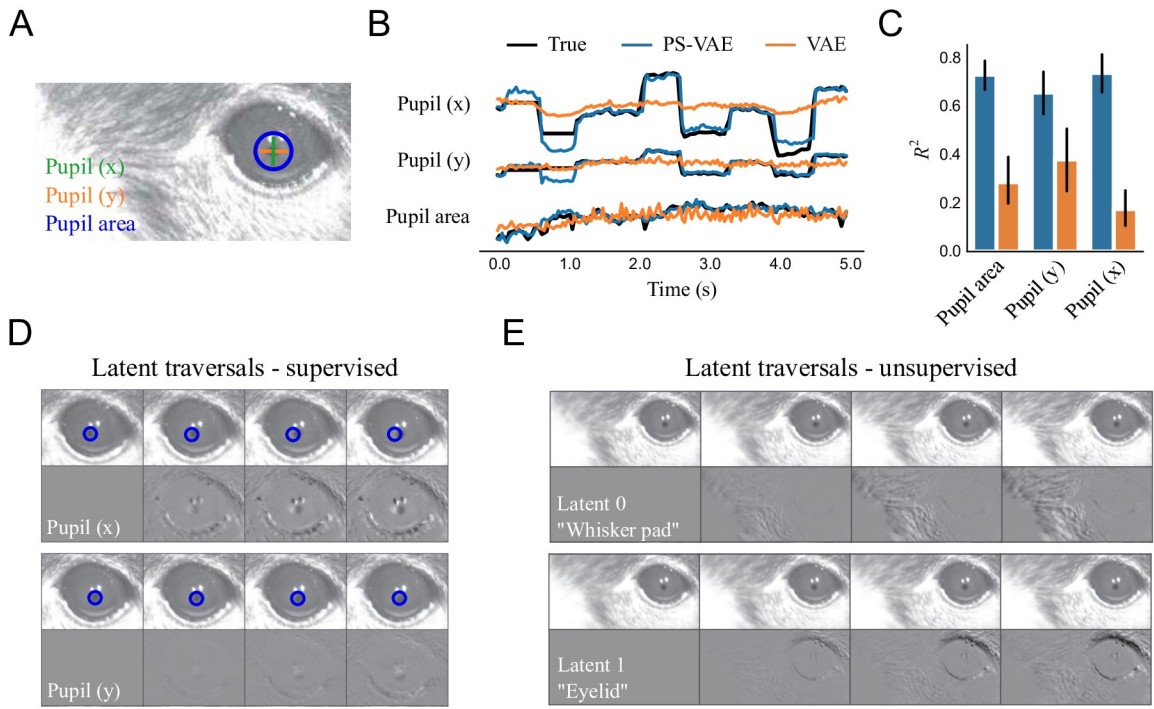

**Fig 4. The PS-VAE successfully partitions the latent representation of a mouse face video. A**: Example frame from the video. Pupil area and pupil location are tracked to provide labels for the PS-VAE supervised subspace. **B**: The true labels (black lines) are again almost perfectly reconstructed by the supervised subspace of the PS-VAE (blue lines). Reconstructions from a standard VAE (orange lines) are able to capture pupil area but miss much of the variability in the pupil location. **C**: Observations from the example trial hold across all labels and test trials. Error bars are computed as in Fig 2D. **D**: Frames generated by manipulating the representation in the supervised subspace. *Top*: Manipulation of the *x* coordinate of the pupil location. The change is slight due to a small dynamic range of the pupil position in the video, so a static blue circle is superimposed as a reference point. *Bottom*: Manipulation of the *y* coordinate of the pupil location. **E**: Same as panel D except the manipulation is performed in the two unsupervised dimensions. Latent 0 encodes the position of the whisker pad, while Latent 1 encodes the position of the eyelid. See S14 Video for a dynamic version of these traversals.

with their corresponding PCA signals (whisker pad, $r = 0.51$; eyelid $r = 0.92$; S1 Fig). Together these results from the label reconstruction analysis and the latent traversals demonstrate the PS-VAE is able to learn an interpretable representation for this behavioral video.

The separation of eye and whisker pad information allows us to independently characterize the dynamics of each of these behavioral features. As an example of this approach we fit a simple movement detector using a 2-state autoregressive hidden Markov model (ARHMM) [51]. The ARHMM clusters time series data based on dynamics, and we typically find that a 2-state ARHMM clusters time points into "still" and "moving" states of the observations [13, 27]. We first fit the ARHMM on the pupil location latents, where the "still" state corresponds to periods of fixation, and the "moving" state corresponds to periods of pupil movement; the result is a saccade detector (Fig 5A). Indeed, if we align all the PS-VAE latents to saccade onsets found by the ARHMM, we find variability in the pupil location latents increases just after the saccades (Fig 5C). See S21 Video for example saccade clips, and S16 Fig for panel captions. This saccade detector could have been constructed using the original pupil location labels, so we next fit the ARHMM on the whisker pad latents, obtained from the unsupervised latents, which results in a whisker pad movement detector (Fig 5B and 5D; see S22 Video for example movements). To validate this detector we compare it to a hand engineered detector, in which we fit an ARHMM to the 1D whisker signal extracted from the whisker pad; these hand engineered states have 87.1% overlap with the states from the PS-VAE-based detector (S1 Fig). The

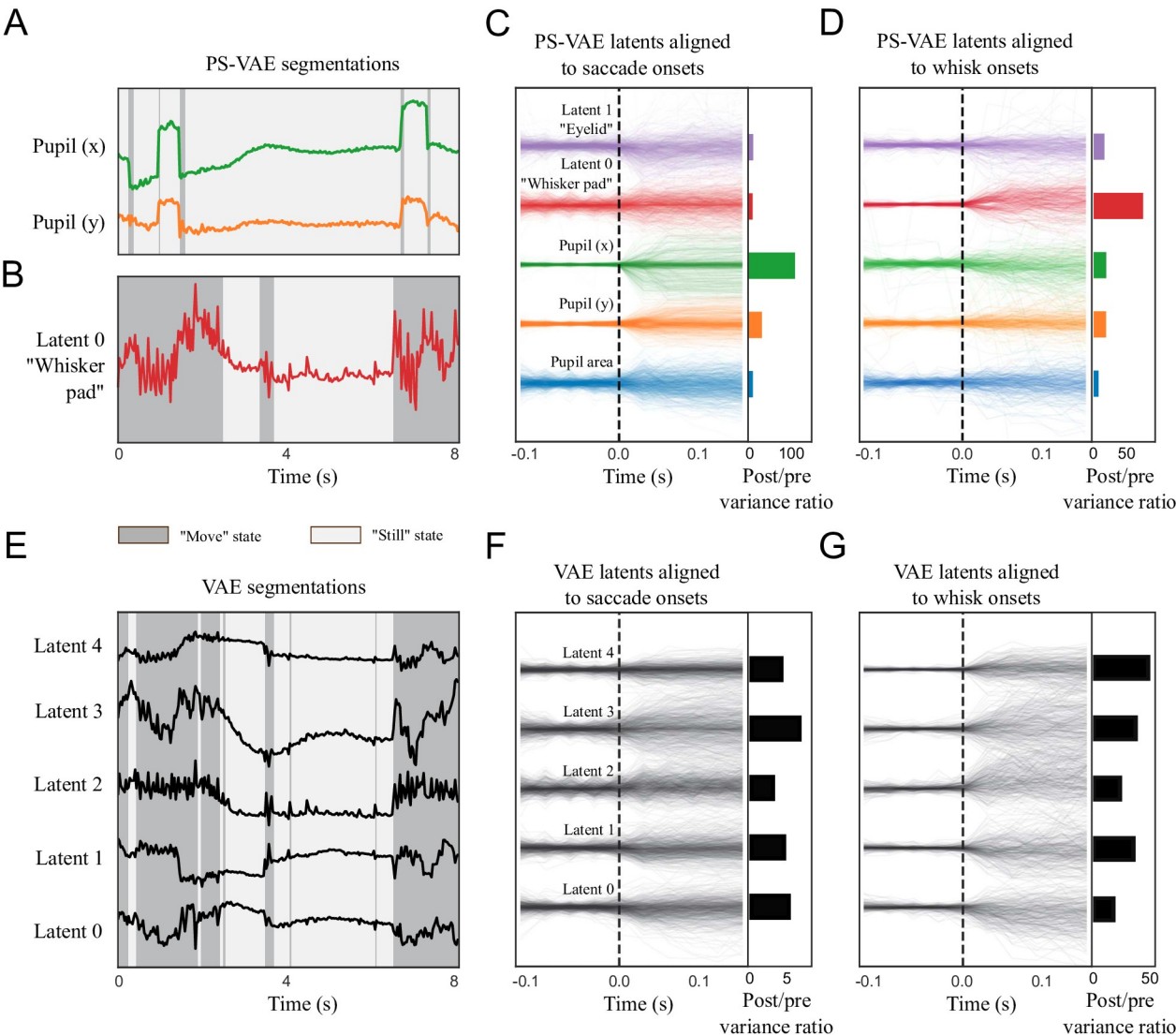

**Fig 5. The PS-VAE enables targeted downstream behavioral analyses of the mouse face video.** A simple 2-state autoregressive hidden Markov model (ARHMM) is used to segment subsets of latents into "still" and "moving" states (which refer only to the behavioral features modeled by the ARHMM, not the overall behavioral state of the mouse). **A**: An ARHMM is fit to the two supervised latents corresponding to the pupil location, resulting in a saccade detector (S21 Video). Background colors indicate the most likely state at each time point. **B**: An ARHMM is fit to the single unsupervised latent corresponding to the whisker pad location, resulting in a whisking detector (S22 Video). **C**: *Left*: PS-VAE latents aligned to saccade onsets found by the model from panel A. *Right*: The ratio of post-saccade to pre-saccade activity shows the pupil location has larger modulation than the other latents. **D**: PS-VAE latents aligned to onset of whisker pad movement; the largest increase in variability is seen in the whisker pad latent. **E**: An ARHMM is fit to five fully unsupervised latents from a standard VAE. The ARHMM can still reliably segment the traces into "still" and "moving" periods, although these tend to align more with movements of the whisker pad than the pupil location (compare to segmentations in panels A and B). **F**: VAE latents aligned to saccade onsets found by the model from panel A. Variability after saccade onset increases across many latents, demonstrating the distributed nature of the pupil location representation. **G**: VAE latents aligned to whisker movement onsets found by the model from panel B. The whisker pad is clearly represented across all latents. This distributed representation makes it difficult to interpret individual VAE latents, and therefore does not allow for the targeted behavioral models enabled by the PS-VAE.

interpretable PS-VAE latents thus allow us to easily fit several simple ARHMMs to different behavioral features, rather than a single complex ARHMM (with more states) to all behavioral features. This is a major advantage of the PS-VAE framework, because we find that ARHMMs provide more reliable and interpretable output when used with a small number of states, both in simulated data (S2 Fig) and in this particular dataset (S3 Fig).

We now repeat the above analysis with the latents of a VAE to further demonstrate the advantage gained by using the PS-VAE in this behavioral analysis. We fit a 2-state ARHMM to all latents of the VAE (since we cannot easily separate different dimensions) and again find "still" and "moving" states, which are highly overlapping with the whisker pad states found with the PS-VAE (92.2% overlap) and the hand engineered whisker pad states (85.0% overlap). However, using the VAE latents, we are not able to easily discern the pupil movements (70.2% overlap). This is due in part to the fact that the VAE latents do not contain as much pupil information as the PS-VAE (Fig 4C), and also due to the fact that what pupil information does exist is generally masked by the more frequent movements of the whisker pad (Fig 5A and 5B). Indeed, plotting the VAE latents aligned to whisker pad movement onsets (found from the PS-VAE-based ARHMM) shows a robust detection of movement (Fig 5G), and also shows that the whisker pad is represented non-specifically across all VAE latents. However, if we plot the VAE latents aligned to saccade onsets (found from the PS-VAE-based ARHMM), we also find variability after saccade onset increases across all latents (Fig 5F). So although the VAE movement detector at first seems to mostly capture whisker movements, it is also contaminated by eye movements.

A possible solution to this problem is to increase the number of ARHMM states, so that the model may find different combinations of eye movements and whisker movements (i.e. eye still/whisker still, eye moving/whisker still, etc.). To test this we fit a 4-state ARHMM to the VAE latents, but find the resulting states do not resemble those inferred by the saccade and whisking detectors, and in fact produce a much noisier segmentation than the combination of simpler 2-state ARHMMs (S3 Fig). Therefore we conclude that the entangled representation of the VAE does not allow us to easily construct saccade or whisker pad movement detectors, as does the interpretable representation of the PS-VAE.

The separation of eye and whisker pad information also allows us to individually decode these behavioral features from neural activity. In this dataset, neural activity in primary visual cortex was optically recorded using two-photon calcium imaging. We randomly subsample 200 of the 1370 recorded neurons and decode the PS-VAE latents using nonlinear MLP regression (Fig 6A and 6B). We repeated this subsampling process 10 times, and find that the neural activity is able to successfully reconstruct the pupil area, eyelid, and horizontal position of the pupil location, but does not perform as well reconstructing the whisker pad or the vertical position of the pupil location (which may be due to the small dynamic range of the vertical position and the accompanying noise in the labels) (Fig 6C). Furthermore, we find these $R^2$ values to be very similar whether decoding the PS-VAE supervised latents (shown here in Fig 6) or the original labels (S8 Fig).

In addition to decoding the PS-VAE latents, we also decoded the motion energy (ME) of the latents (S6 Fig), as previous work has demonstrated that video ME can be an important predictor of neural activity [22, 23, 52]. We find in this dataset that the motion energy of the whisker pad is decoded reasonably well ($R^2 = 0.33 \pm 0.01$), consistent with the results in [22] and [23] that use encoding (rather than decoding) models. The motion energies of the remaining latents (pupil area and location, and eyelid) are not decoded well.

Again we can easily demonstrate the advantage gained by using the PS-VAE in this analysis by decoding the VAE latents (Fig 6D). We find that one latent dimension in particular is decoded well (Fig 6E, Latent 4). Upon reviewing the latent traversal video for this VAE (S15 Video), we find that Latent 4 encodes information about every previously described behavioral feature—pupil location, pupil area, whisker pad, and eyelid. This entangled VAE representation makes it difficult to understand precisely how well each of those behavioral features is represented in the neural activity; the specificity of the PS-VAE behavioral representation, on the other hand, allows for a greater specificity in neural decoding.

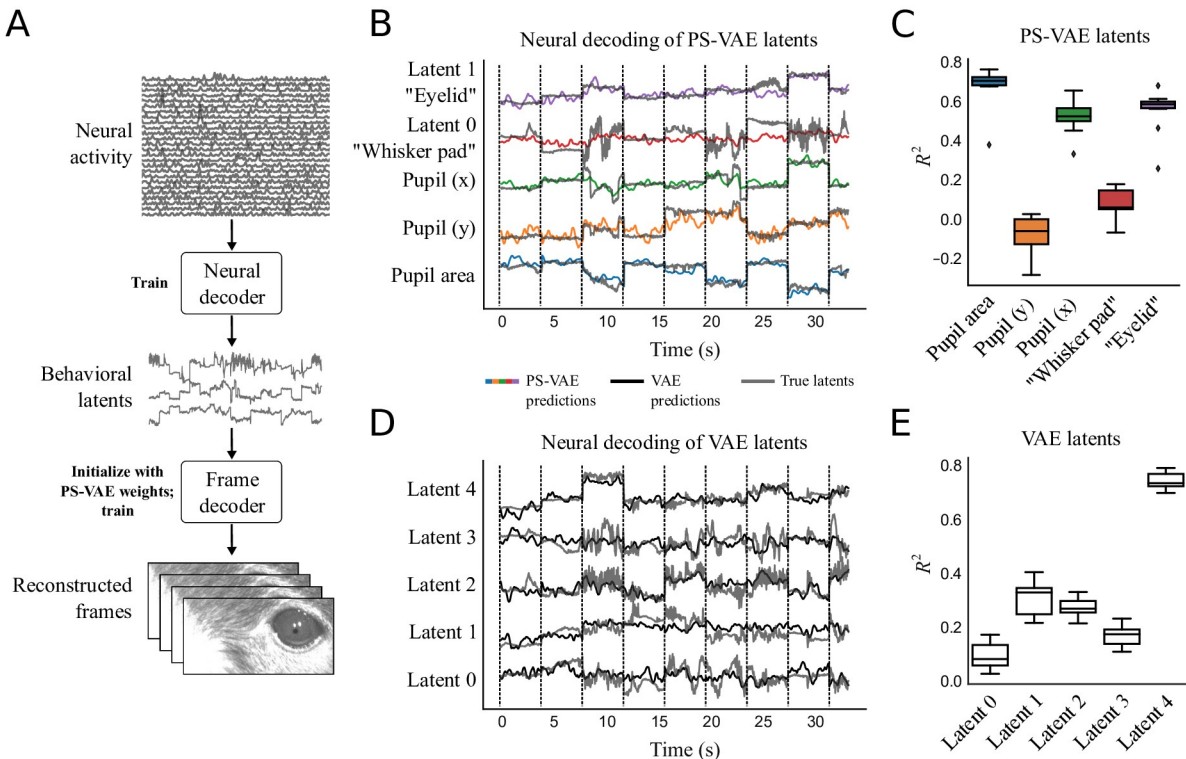

**Fig 6. The PS-VAE enables targeted downstream neural analyses of the mouse face video. A**: A neural decoder is trained to map neural activity to the interpretable behavioral latents. These predicted latents can then be further mapped through the frame decoder learned by the PS-VAE to produce video frames reconstructed from neural activity. **B**: PS-VAE latents (gray traces) and their predictions from neural activity (colored traces) recorded in primary visual cortex with two-photon imaging. Vertical black lines delineate individual test trials. See S25 Video for a video of the full frame decoding. **C**: Decoding accuracy ($R^2$) computed separately for each latent demonstrates how the PS-VAE can be utilized to investigate the neural representation of different behavioral features. Boxplots show variability over 10 random subsamples of 200 neurons from the full population of 1370 neurons. **D**: Standard VAE latents (gray traces) and their predictions from the same neural activity (black traces). **E**: Decoding accuracy for each VAE dimension reveals one dimension that is much better decoded than the rest, but the distributed nature of the VAE representation makes it difficult to understand which behavioral features the neural activity is predicting.

We can take this decoding analysis one step further and decode not only the behavioral latents, but the behavioral videos themselves from neural activity. To do so we retrain the PS-VAE's convolutional decoder to map from the neural predictions of the latents (rather than the latents themselves) to the corresponding video frame (Fig 6A). The result is an animal behavioral video that is fully reconstructed from neural activity. See S25 Video for the neural reconstruction video, and S17 Fig for panel captions. These videos can be useful for gaining a qualitative understanding of which behavioral features are captured (or not) by the neural activity—for example, it is easy to see in the video that the neural reconstruction typically misses high frequency movements of the whisker pad. It is also possible to make these reconstructed videos with the neural decoder trained on the VAE latents and the corresponding VAE frame decoder [27]. These VAE reconstructions are qualitatively and quantitatively similar to the PS-VAE reconstructions, suggesting the PS-VAE can provide interpretability without sacrificing information about the original frames in the latent representation.

**A two-view mouse video.** The next dataset that we consider [22] poses a different set of challenges than the previous datasets. This dataset uses two cameras to simultaneously capture the face and body of a head-fixed mouse in profile and from below (Fig 7A). Notably, the cameras also capture the movements of two lick spouts and two levers. As we show later, the

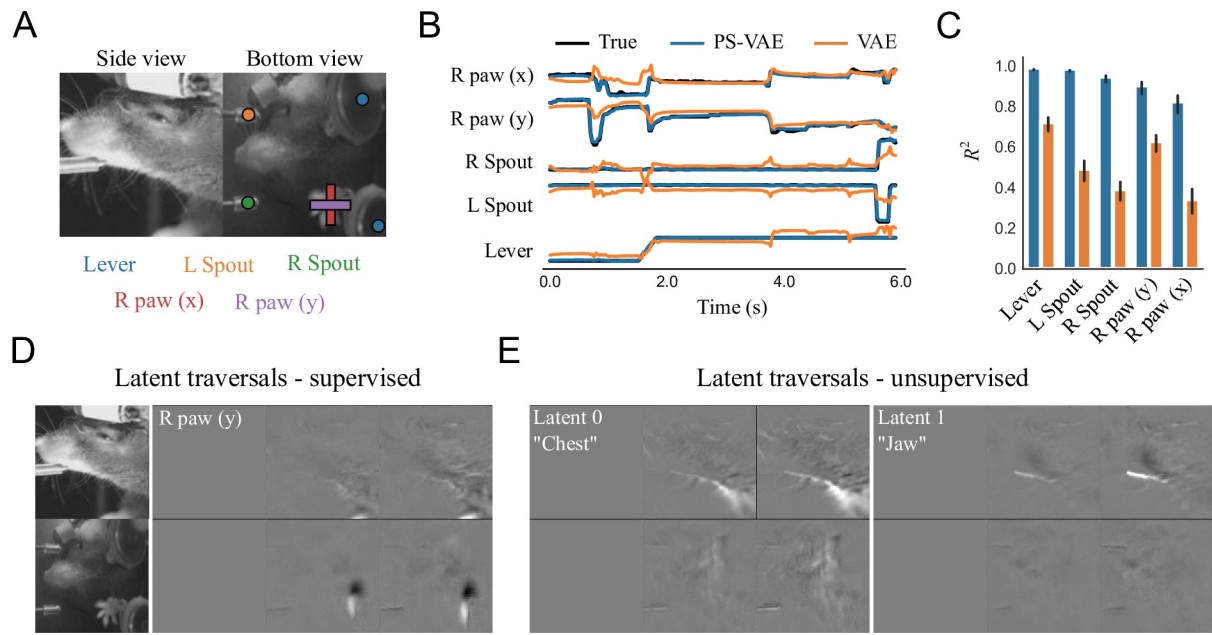

**Fig 7. The PS-VAE successfully partitions the latent representation of a two-view mouse video [22].** **A**: Example frames from the video. Mechanical equipment (lever and two independent spouts) as well as the single visible paw are tracked to provide labels for the PS-VAE supervised subspace. By tracking the moving mechanical equipment, the PS-VAE can isolate this variability in a subset of the latent dimensions, allowing the remaining dimensions to solely capture the animal's behavior. **B**: The true labels (black lines) are again almost perfectly reconstructed by the supervised subspace of the PS-VAE (blue lines). Reconstructions from a standard VAE (orange lines) miss much of the variability in these labels. **C**: Observations from the example trial hold across all labels and test trials. Error bars are computed as in Fig 2D. **D**: Frames generated by manipulating the $y$ coordinate of the tracked paw results in changes in the paw position, and only small changes in the side view. Only differenced frames are shown for clarity. **E**: Manipulation of the two unsupervised dimensions. Latent 0 (*left*) encodes the position of the chest, while Latent 1 (*right*) encodes the position of the jaw. The contrast of the latent traversal frames has been increased for visual clarity. See S16 Video for a dynamic version of these traversals.

movement of this mechanical equipment drives a significant fraction of the pixel variance, and is thus clearly encoded in the latent space of the VAE. By tracking this equipment we are able to encode mechanical movements in the supervised subspace of the PS-VAE, which allows the unsupervised subspace to capture solely animal-related movements.

We tracked the two moving lick spouts, two moving levers, and the single visible paw using DeepLabCut [10]. The lick spouts move independently, but only along a single dimension, so we were able to use one label (i.e. one dimension) for each spout. The levers always move synchronously, and only along a one-dimensional path, so we were able to use a single label for all lever-related movement. Therefore in our analysis we use models with a 7D latent space: a 5D supervised subspace (three equipment labels plus the $x$, $y$ coordinates of the visible paw) and a 2D unsupervised subspace. To incorporate the two camera views into the model we resized the frames to have the same dimensions, then treated each grayscale view as a separate channel (similar to having separate red, green, and blue channels in an RGB image).

The PS-VAE is able to successfully reconstruct all of the labels ($R^2 = 0.93 \pm 0.01$), again outperforming the linear regression from the VAE latents ($R^2 = 0.53 \pm 0.01$) (Fig 7B and 7C) as well as the nonlinear MLP regression from the VAE latents ($R^2 = 0.85 \pm 0.01$). The latent traversals in the supervised subspace also show the PS-VAE learned to capture the label information (Fig 7D). The latent traversals in the unsupervised subspace show one dimension related to the chest and one dimension related to the jaw location (Fig 7E), two body parts that are otherwise hard to manually label (S16 Video). Nevertheless, we attempted to extract 1D hand

engineered signals from crops of the chest and jaw as in the mouse face dataset, and again find reasonable correlation with the corresponding PS-VAE latents (chest, $r$ = 0.41; jaw, $r$ = 0.76; S4 Fig). [We note that an analogue of the chest latent is particularly difficult to extract from the frames by cropping due to the presence of the levers and paws.] Together these results from the label reconstruction analysis and the latent traversals demonstrate that, even with two concatenated camera views, the PS-VAE is able to learn an interpretable representation for this behavioral video.

We also use this dataset to demonstrate that the PS-VAE can find more than two interpretable unsupervised latents. We removed the paw labels and refit the PS-VAE with a 3D supervised subspace (one dimension for each of the equipment labels) and a 4D unsupervised subspace. We find that this model recovers the original unsupervised latents—one for the chest and one for the jaw—and the remaining two unsupervised latents capture the position of the (now unlabeled) paw, although they do not learn to strictly encode the $x$ and $y$ coordinates (see S17 Video; "R paw 0" and "R paw 1" panels correspond to the now-unsupervised paw dimensions).

As previously mentioned, one major benefit of the PS-VAE for this dataset is that it allows us find a latent representation that separates the movement of mechanical equipment from the movement of the animal. To demonstrate this point we align the PS-VAE and VAE latents to the time point where the levers move in for each trial (Fig 8A). The PS-VAE latent corresponding to the lever increases with little trial-to-trial variability (blue lines), while the animal-related latents show extensive trial-to-trial variability. On the other hand, the VAE latents show activity that is locked to lever movement onset across many of the dimensions, but it is not straightforward to disentangle the lever movements from the body movements here. The PS-VAE thus provides a substantial advantage over the VAE for any experimental setup that involves moving mechanical equipment that is straightforward to track.

Beyond separating equipment- and animal-related information, the PS-VAE also allows us to separate paw movements from body movements (which we take to include the jaw). As in the mouse face dataset, we demonstrate how this separation allows us to fit some simple movement detectors to specific behavioral features. We fit 2-state ARHMMs separately on the paw latents (Fig 8B) and the body latents (Fig 8C) from the PS-VAE. To validate the body movement detector—constructed from PS-VAE unsupervised latents—we fit another ARHMM to the 1D body signal analogues extracted directly from the video frames, and find high overlap (94.9%) between the states of these two models (S4 Fig). We also fit an ARHMM on all latents from the VAE (Fig 8D). Again we see the VAE segmentation tends to line up with one of these more specific detectors more than the other (VAE and PS-VAE-paw state overlap: 72.5%; VAE and PS-VAE-body state overlap: 95.3%; VAE and hand engineered body state overlap: 93.1%). If we align all the PS-VAE latents to paw movement onsets found by the ARHMM (Fig 8E, *top*), we can make the additional observation that these paw movements tend to accompany body movements, as well as lever movements (see S23 Video for example clips). However, this would be impossible to ascertain from the VAE latents alone (Fig 8E, *bottom*), where the location of the mechanical equipment, the paw, and the body are all entangled. We make a similar conclusion when aligning the latents to body movement onsets (Fig 8F; see S24 Video for example clips). Furthermore, we find that increasing the number of ARHMM states does not help with interpretability of the VAE states (S5 Fig). The entangled representation of the VAE therefore does not allow us to easily construct paw or body movement detectors, as does the interpretable representation of the PS-VAE.

Finally, we decode the PS-VAE latents—both equipment- and animal-related—from neural activity. In this dataset, neural activity across dorsal cortex was optically recorded using widefield calcium imaging. We extract interpretable dimensions of neural activity using LocaNMF

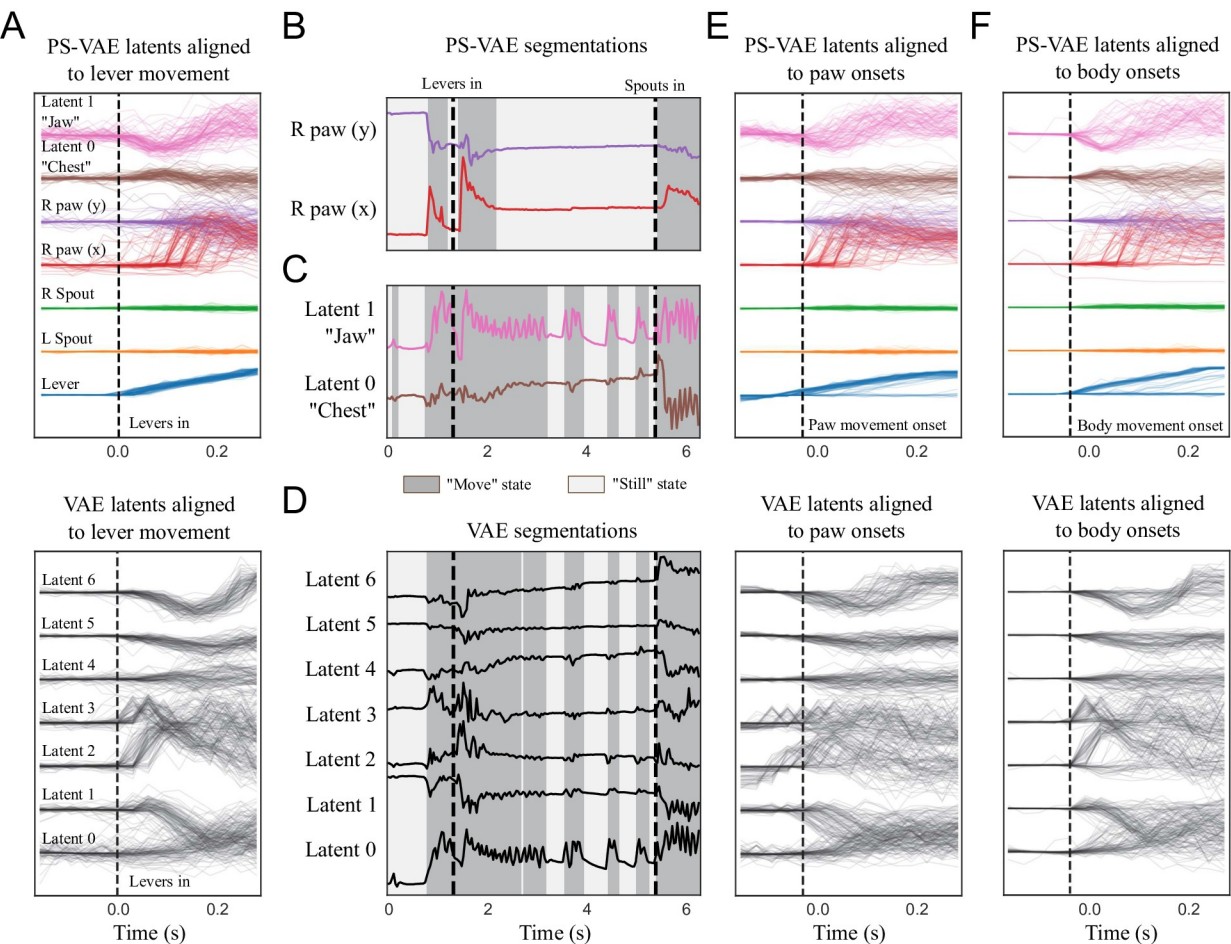

**Fig 8. The PS-VAE enables targeted downstream behavioral analyses of the two-view mouse video. A**: PS-VAE latents (*top*) and VAE latents (*bottom*) aligned to the lever movement. The PS-VAE isolates this movement in the first (blue) dimension, and variability in the remaining dimensions is behavioral rather than mechanical. The VAE does not clearly isolate the lever movement, and as a result it is difficult to distinguish variability that is mechanical versus behavioral. **B**: An ARHMM is fit to the two supervised latents corresponding to the paw position (S23 Video). Background colors as in Fig 5. **C**: An ARHMM is fit to the two unsupervised latents corresponding to the chest and jaw, resulting in a "body" movement detector that is independent of the paw (S24 Video). **D**: An ARHMM is fit to seven fully unsupervised latents from a standard VAE. The "still" and "moving" periods tend to align more with movements of the body than the paw (compare to panels B and C). **E**: PS-VAE latents (*top*) and VAE latents (*bottom*) aligned to the onsets of paw movement found in B. This movement also is often accompanied by movements of the jaw and chest, although this is impossible to ascertain from the VAE latents. **F**: This same conclusion holds when aligning the latents to the onsets of body movement.

[19], which finds a low-dimensional representation for each of 12 aggregate brain regions defined by the Allen Common Coordinate Framework Atlas [53]. We first decode the PS-VAE latents from all brain regions using nonlinear MLP regression and find good reconstructions (Fig 9A), even for the equipment-related latents (quantified in S7 Fig). The real benefit of our approach becomes clear, however, when we perform a region-based decoding analysis (Fig 9B). The confluence of interpretable, region-based neural activity with interpretable behavioral latents from the PS-VAE leads to a detailed mapping between specific brain regions and specific behaviors.

In this detailed mapping we see the equipment-related latents actually have higher reconstruction quality than the animal-related latents, although the equipment-related latents contain far less trial-to-trial variability (Fig 8A). Of the animal-related latents, the *x* coordinate of

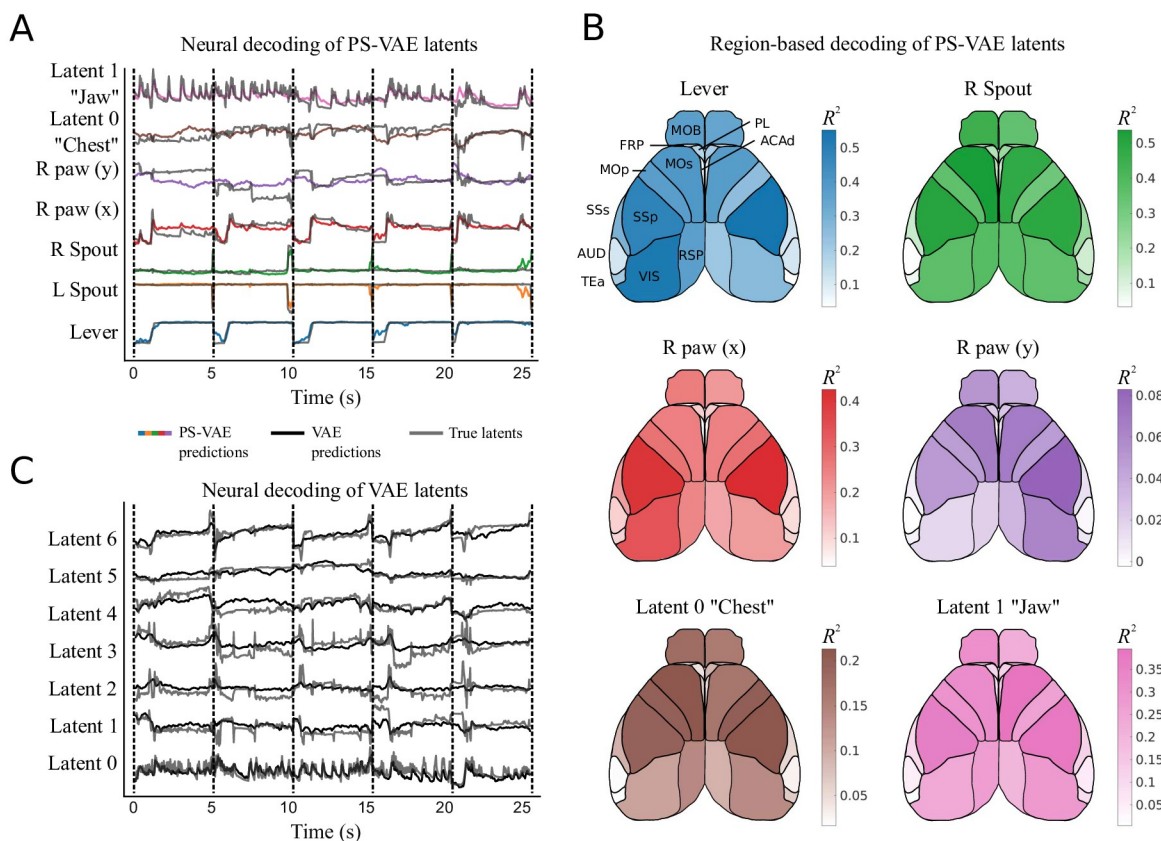

**Fig 9. The PS-VAE enables a detailed brain region-to-behavior mapping in the two-view mouse dataset. A**: PS-VAE latents (gray traces) and their predictions from neural activity (colored traces) recorded across dorsal cortex with widefield calcium imaging. Vertical dashed black lines delineate individual test trials. See S26 Video for a video of the full frame decoding. **B**: The behavioral specificity of the PS-VAE can be combined with the anatomical specificity of computational tools like LocaNMF [19] to produce detailed mappings from distinct neural populations to distinct behavioral features. Region acronyms are defined in Table 1. **C**: VAE latents (gray traces) and their predictions from the same neural activity as in A (black traces). The distributed behavioral representation produced by the VAE does not allow for the same region-to-behavior mappings enabled by the PS-VAE.

the right paw (supervised) and the jaw position (unsupervised) are the best reconstructed, followed by the chest and then the *y* coordinate of the right paw. Most of the decoding power comes from the motor (MOp, MOs) and somatosensory (SSp, SSs) regions, although visual regions (VIS) also perform reasonably well. We note that, while we could perform a region-based decoding of VAE latents [27], the lack of interpretability of those latents does not allow for the same specificity as the PS-VAE.

## Extending the PS-VAE framework to multiple sessions

Thus far we have focused on validating the PS-VAE on single videos from different experimental setups; in practice, though, we will typically want to produce a low-dimensional latent representation that is shared across multiple experimental sessions, rather than fitting session-specific models. However, the inclusion of multiple videos during training introduces a new problem: different videos from the same experimental setup will contain variability in the experimental equipment, lighting, or even physical differences between animals, despite efforts to standardize these features (Fig 10A). We do not want these differences (which we refer to

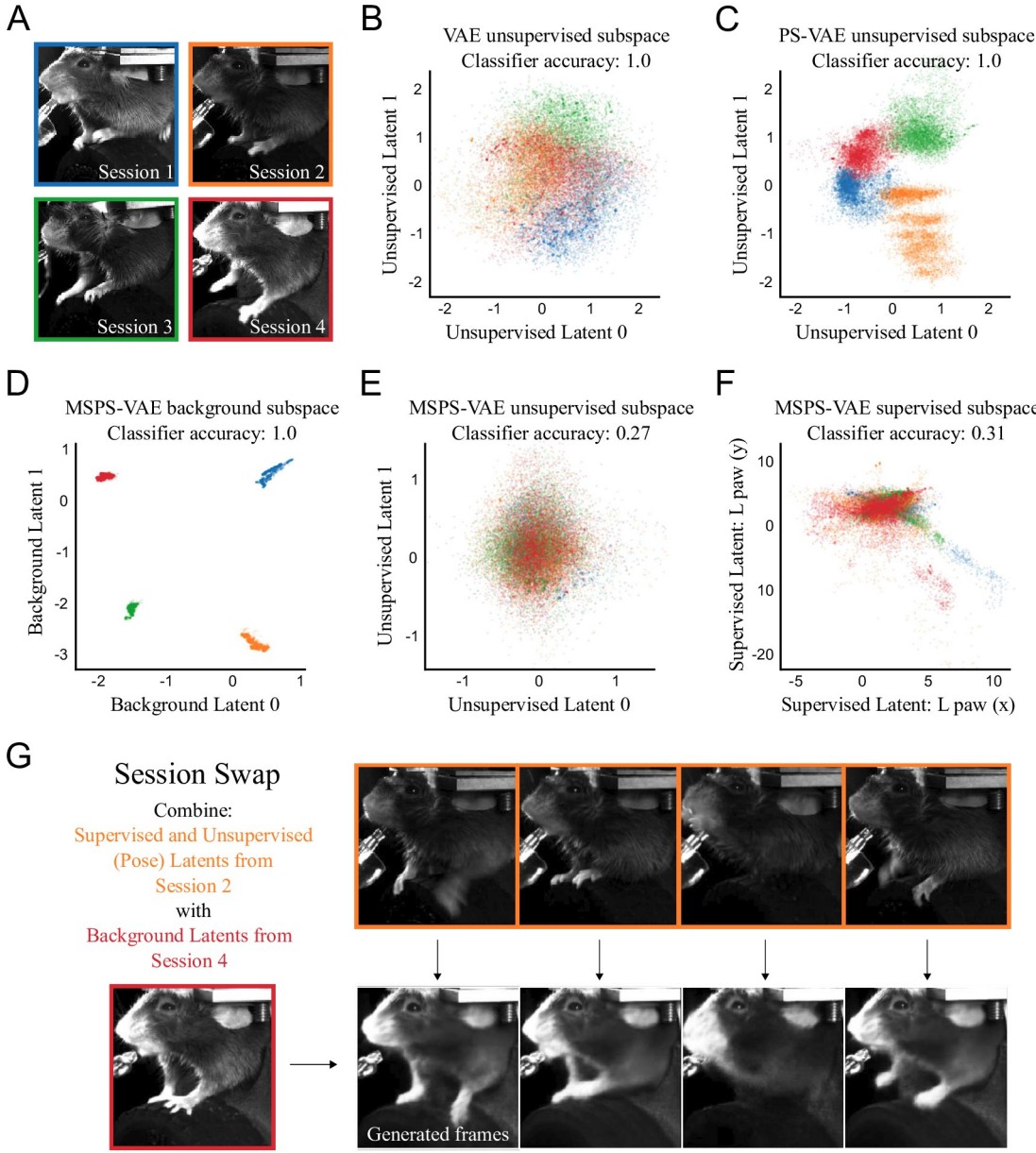

**Fig 10. The multi-session PS-VAE (MSPS-VAE) accounts for session-level differences between videos in the head-fixed mouse dataset [46]. A**: One example frame from each of four experimental sessions with variation in lighting, experimental equipment, and animal appearance. **B**: Distribution of two latents from a VAE trained on all four sessions. Noticeable session-related structure is present, and a linear classifier can perfectly predict session identity on held-out test data (note the VAE has a total of eleven latent dimensions). Colors correspond to borders in panel A. **C**: Distribution of two unsupervised latents from a PS-VAE. **D**: Distribution of two background latents from an MSPS-VAE, which are designed to contain all of the static across-session variability. **E**: Distribution of two unsupervised latents from an MSPS-VAE. Note the lack of session-related structure; a linear classifier can only predict 27% of the data points correctly (chance level is 25%). **F**: Distribution of two supervised latents from an MSPS-VAE. **G**: Example of a "session swap" where the pose of one mouse is combined with the background appearance of another mouse to generate new frames. These swaps qualitatively demonstrate the model has learned to successfully encode these different features in the proper subspaces. See S27 Video for a dynamic version of these swaps.

collectively as the "background") to contaminate the latent representation, as they do not contain the behavioral information we wish to extract for downstream analyses.

To demonstrate how the background can affect the latent representation, we first fit a standard VAE using four sessions from the same experimental setup as the head-fixed mouse dataset (Fig 10A; see S1 Table for hyperparameter settings for this and subsequent models). The distribution of latents shows considerable session-specific structure (Fig 10B). We quantify this observation by fitting a logistic regression classifier from the VAE latent space to the session identity for each time point. We find this classifier can predict the correct session on held-out test data 100% of the time. However, a classifier fit to the z-scored labels predicts the correct session only 22% of the time (chance level is 25%). This demonstrates that much of the information concerning session identity comes not from the labels (i.e. the behavior), but instead from the background information. The standard PS-VAE does not perform any better; a classifier fit to the PS-VAE unsupervised latents can again perfectly predict session identity (Fig 10C).

To address this issue within the framework of the PS-VAE, we introduce a new subspace into our model which captures static differences between sessions (the "background" subspace) while leaving the other subspaces (supervised and unsupervised) to capture dynamic behaviors. As before, we briefly describe this multi-session PS-VAE (MSPS-VAE) here, while full details can be found in the Methods. Recall that the PS-VAE forms the supervised and unsupervised latent representations $\mathbf{z}_s$ and $\mathbf{z}_u$ as a nonlinear transformation of the observed video frame $\mathbf{x}$. In a similar fashion, we define the background representation $\mathbf{z}_b$ and encourage all three subspaces to be orthogonal to one another. We employ a triplet loss [54] that moves data points from the same session nearer to each other in the background subspace, and pushes data points from different sessions farther apart. Finally, the concatenated latent representation $\mathbf{z} = [\mathbf{z}_u; \mathbf{z}_s; \mathbf{z}_b]$ is used to reconstruct the observed video frame.

We fit the MSPS-VAE to the four sessions in Fig 10A, and visualize the three latent spaces. The background subspace, trained with the triplet loss, has learned to cleanly separate the four sessions (Fig 10D). An immediate consequence of this separation is that the distribution of data points in the unsupervised subspace no longer contains session-specific structure (Fig 10E); in fact, a classifier can only predict the correct session identity 27% of the time (where again chance level is 25%). We also find these dimensions to be relatively interpretable, and conserved across the four different sessions (S20 Video). Finally, we see little session-specific structure in the supervised latents (Fig 10F), and note that label reconstructions are still good and outperform VAE label reconstructions (S9 Fig).

Once the MSPS-VAE has been trained, the background representation $\mathbf{z}_b$ should be independent of the behavioral information represented in $\mathbf{z}_s$ and $\mathbf{z}_u$. As a qualitative test of this, we perform a "session swap" where we combine the poses of one session (encoded in $\mathbf{z}_s$ and $\mathbf{z}_u$) with the appearance of another session (encoded in $\mathbf{z}_b$). This allows us to, for example, take a clip from one session, then generate corresponding clips that appear as if animals from the other sessions are performing the exact same movements (Fig 10G; see S27 Video, and S18 Fig for panel captions). We find that the movements of obvious body parts such as the paws and tongue are synchronized across the videos, while background information such as the lighting, lickspout, and head stage acquire the appearance of the desired session.

## Discussion

In this work we introduced the Partitioned Subspace VAE (PS-VAE), a model that produces interpretable, low-dimensional representations of behavioral videos. We applied the PS-VAE to three head-fixed mouse datasets (Figs 2, 4 and 7) and a freely moving mouse dataset (Fig 3),

demonstrating on each that our model is able to extract a set of supervised latents corresponding to user-supplied labels, and another set of unsupervised latents that account for other salient behavioral features. Notably, the PS-VAE can accommodate a range of tracking algorithms—the analyzed datasets contain labels from Deep Graph Pose [13] (head-fixed mouse), Facemap [50] (mouse face), and DeepLabCut [10] (freely moving mouse, two-view mouse). [Although all of our examples use pose estimates as labels, this is not an explicit requirement of the model; the labels can in general be any variable, continuous or discrete, that might be predicted from the video data.] We then demonstrated how the PS-VAE's interpretable representations lend themselves to targeted downstream analyses which were otherwise infeasible using supervised or unsupervised methods alone. In one dataset we constructed a saccade detector from the supervised representation, and a whisker pad movement detector from the unsupervised representation (Fig 5); in a second dataset we constructed a paw movement detector from the supervised representation, and a body movement detector from the unsupervised representation (Fig 8). We then decoded the PS-VAE's behavioral representations from neural activity, and showed how their interpretability allows us to better understand how different brain regions are related to distinct behaviors. For example, in one dataset we found that neurons from visual cortex were able to decode pupil information much more accurately than whisker pad position (Fig 6); in a second dataset we separately decoded mechanical equipment, body position, and paw position from across the dorsal cortex (Fig 9). Finally, we extended the PS-VAE framework to accommodate multiple videos from the same experimental setup (Fig 10). To do so we introduced a new subspace that captures variability in static background features across videos, while leaving the original subspaces (supervised and unsupervised) to capture dynamic behavioral features.

The PS-VAE contributes to a growing body of research that relies on automated video analysis to facilitate scientific discovery, which often requires supervised or unsupervised dimensionality reduction approaches to first extract meaningful behavioral features from video. Notable examples include "behavioral phenotyping," a process which can automatically compare animal behavior across different genetic populations, disease conditions, and pharmacological interventions [16, 55]; the study of social interactions [56–59]; and quantitative measurements of pain response [60] and emotion [61]. The more detailed behavioral representation provided by the PS-VAE enables future such studies to consider a wider range of behavioral features, potentially offering a more nuanced understanding of how different behaviors are affected by genes, drugs, and the environment.

Automated video analysis is also becoming central to the search for neural correlates of behavior. Several recent studies applied PCA (an unsupervised approach) to behavioral videos to demonstrate that movements are encoded across the entire mouse brain, including regions not previously thought to be motor-related [22, 23]. In contrast to PCA, which does not take into account external covariates, the PS-VAE extracts interpretable pose information, as well as automatically discovers additional sources of variation in the video. These interpretable behavioral representations, as shown in our results (Figs 6 and 9), lead to more refined correlations between specific behaviors and specific neural populations. Moreover, motor control studies have employed supervised pose estimation algorithms to extract kinematic quantities and regress them against simultaneously recorded neural activity [56, 62–65]. The PS-VAE may allow such studies to account for movements that are not easily captured by tracked key points, such as soft tissues (e.g. a whisker pad or throat) or body parts that are occluded (e.g. by fur or feathers).

Finally, an important thread of work scrutinizes the neural underpinnings of naturalistic behaviors such as rearing [25] or mounting [66]. These discrete behaviors are often extracted from video data via segmentation of a low-dimensional representation (either supervised or

unsupervised), as we demonstrated with the ARHMMs (Figs 5 and 8). Here too, the interpretable representation of the PS-VAE can allow segmentation algorithms to take advantage of a wider array of interpretable features, producing a more refined set of discrete behaviors.

In the results presented here we have implicitly defined "interpretable" behavioral features to mean individual body parts such as whisker pads, eyelids, and jaws. However, we acknowledge that "interpretable" is a subjective term [67], and will carry different meanings for different datasets. It is of course possible that the PS-VAE could find "interpretable" features that involve multiple coordinated body parts. Furthermore, features that are not immediately interpretable to a human observer may still contain information that is relevant to the scientific question at hand. For example, when comparing the behaviors of two subject cohorts (e.g. healthy and diseased) we might find that a previously uninterpretable feature is a significant predictor of cohort. Regardless of whether or not the unsupervised latents of the PS-VAE map onto intuitive behavioral features, these latents will still account for variance that is not explained by the user-provided labels.

There are some obvious directions to explore by applying the PS-VAE to different species and different experimental preparations, though the model may not be appropriate for analyzing behavioral videos where tracking non-animal equipment is not possible. Examples include bedding that moves around in a home cage experiment, or a patterned ball used for locomotion [68]. Depending on the amount of pixel variance driven by changes in these non-trackable, non-behavioral features, the PS-VAE may attempt to encode them in its unsupervised latent space. This encoding may be difficult to control, and could lead to uninterpretable latents.

The PS-VAE is not limited to the analysis of video data; rather, it is a general purpose nonlinear dimensionality reduction tool that partitions the low-dimensional representation into a set of dimensions that are constrained by user-provided labels, and another set of dimensions that account for remaining variability (similar in spirit to demixed PCA [69]). As such its application to additional types of data is a rich direction for future work. For example, the model could find a low-dimensional representation of neural activity, and constrain the supervised subspace with a low-dimensionsal representation of the behavior—whether that be from pose estimation, a purely behavioral PS-VAE, or even trial variables provided by the experimenter. This approach would then partition neural variability into a behavior-related subspace and a non-behavior subspace. [70] and [71] both propose a linear version of this model, although incorporating the nonlinear transformations of the autoencoder may be beneficial in many cases. [72] take a related nonlinear approach that incorporates behavioral labels differently from our work. Another use case comes from spike sorting: many pipelines contain a spike waveform featurization step, the output of which is used for clustering [73, 74]. The PS-VAE could find a low-dimensional representation of spike waveforms, and constrain the supervised subspace with easy to compute features such as peak-to-peak amplitude and waveform width. The unsupervised latents could then reveal interpretable dimensions of spike waveforms that are important for distinguishing different cells.

The structure of the PS-VAE fuses a generative model of video frames with a discriminative model that predicts the labels from the latent representation [30–37], and we have demonstrated how this structure is able to produce a useful representation of video data (e.g. Fig 2). An alternative approach to incorporating label information is to condition the latent representation directly on the labels, instead of predicting them with a discriminative model [72, 75–82]. We pursued the discriminative (rather than conditional) approach based on the nature of the labels we are likely to encounter in the analysis of behavioral videos, i.e. pose estimates: although pose estimation has rapidly become more accurate and robust, we still expect some degree of noise in the estimates. With the discriminative approach we can

explicitly model that noise with the label likelihood term in the PS-VAE objective function. This approach also allows us to easily incorporate a range of label types beyond pose estimates, both continuous (e.g. running speed or accelerometer data) and discrete (e.g. trial condition or animal identity).

Extending the PS-VAE model itself offers several exciting directions for future work. We note that all of our downstream analyses in this paper first require fitting the PS-VAE, then require fitting a separate model (e.g., an ARHMM, or neural decoder). It is possible to incorporate some of these downstream analyses directly into the model. For example, recent work has combined autoencoders with clustering algorithms [15, 16], similar to what we achieved by separately fitting the ARHMMs (a dynamic clustering method) on the PS-VAE latents. There is also growing interest in directly incorporating dynamics models into the latent spaces of autoencoders for improved video analysis, including Markovian dynamics [83, 84], ARHMMs [26], RNNs [85–89], and Gaussian Processes [90]. There is also room to improve the video frame reconstruction term in the PS-VAE objective function. The current implementation uses the pixel-wise mean square error (MSE) loss. Replacing the MSE loss with a similarity metric that is more tailored to image data could substantially improve the quality of the model reconstructions and latent traversals [88, 91]. And finally, unsupervised disentangling remains an active area of research [38, 42–45, 72, 82, 92, 93], and the PS-VAE can benefit from improvements in this field through the incorporation of new disentangling cost function terms as they become available in the future.

## Methods

### Data details

**Ethics statement.**  Animal experimentation: All procedures and experiments were carried out in accordance with the local laws and following approval by the relevant institutions: the Columbia University Institutional Animal Care and Use Committee [AC-AABL7561] (freely moving mouse dataset) and the Home Office in accordance with the UK Animals (Scientific Procedures) Act 1986 [P1DB285D8] (mouse face dataset).

**Head-fixed mouse dataset [46].**  A head-fixed mouse performed a visual decision-making task by manipulating a wheel with its fore paws. Behavioral data was recorded using a single camera at a 60 Hz frame rate; grayscale video frames were cropped and downsampled to 192x192 pixels. Batches were arbitrarily defined as contiguous blocks of 100 frames. We chose to label the left and right paws (Fig 2) for a total of 4 label dimensions (each paw has an $x$ and $y$ coordinate). For the single session analyzed in Fig 2 we hand labeled 66 frames and trained Deep Graph Pose [13] to obtain labels for the remaining frames. For the multiple sessions analyzed in Fig 10 we used a DLC network [10] trained on many thousands of frames. Each label was individually z-scored to make hyperparameter values more comparable across the different datasets analyzed in this paper, since the label log-likelihood values will depend on the magnitude of the labels. Note, however, that this preprocessing step is not strictly necessary due to the scale and translation transform in Eq 2.

**Freely moving mouse dataset.**  A freely moving mouse explored an open field arena. Behavioral data was recorded using a single overhead camera at a 30 Hz frame rate. We tracked the nose, ears, back, and seven points on the tail with DLC [10], although in our final analysis we dropped all tail labels except the tail base. The video preprocessing (performed after tracking) consisted of background subtraction and egocentric alignment. First, the static background of the original color video was calculated by taking the most frequently occurring RGB values for each pixel. Each frame was compared to this background and those pixels with values within some tolerance level of the background values (75 here) for each RGB channel were

set to white. For egocentric alignment, we calculated the minimum rectangle containing the DLC labels of the left and right ears, back, and tail base. We then cropped and rotated the frame so that the minimum rectangle was centered and aligned, ensuring that the rotation followed a tail to nose orientation from left to right. This alignment fixed the $y$ coordinates of the tail base and left ear, and as such we omitted these labels from further analyses; we individually z-scored each of the remaining labels. Finally, the cropped and rotated frames were converted to grayscale, and have a final size of 192x192 pixels. Batches were arbitrarily defined as contiguous blocks of 128 frames.

**Mouse face dataset.** A head-fixed mouse passively viewed drifting grating stimuli while neural activity in primary visual cortex was optically recorded using two-photon calcium imaging. The mouse was allowed to freely run on a ball. For the acquisition of the face video, two-photon recording, calcium preprocessing, and stimulus presentation we used a protocol similar to [47]. We used a commercial two-photon microscope with a resonant-galvo scanhead (B-scope, ThorLabs, Ely UK), with an acquisition frame rate of about 4.29Hz per plane. We recorded 7 planes with a resolution of 512x512 pixels corresponding to approximately 500 $\mu$m x 500 $\mu$m. Raw calcium movies were preprocessed with Suite2p and transformed into deconvolved traces corresponding to inferred firing rates [94]. Inferred firing rates were then interpolated and sampled to be synchronized with the video camera frames. Videos of the mouse face were captured at 30 Hz with a monochromatic camera while the mouse face was illuminated with a collimated infrared LED. Video frames were spatially downsampled to 256x128 pixels. Batches were arbitrarily defined as contiguous blocks of 150 frames. We used the Facemap software [50] to track pupil location and pupil area (Fig 4), and each of these three labels was individually z-scored.

**Two-view mouse dataset.** [22, 95]. A head-fixed mouse performed a visual decision-making task while neural activity across dorsal cortex was optically recorded using widefield calcium imaging. We used the LocaNMF decomposition approach to extract signals from the calcium imaging video [19, 27] (see Table 1). Behavioral data was recorded using two cameras (one side view and one bottom view; Fig 7) at a 30 Hz framerate, synchronized to the acquisition of neural activity; grayscale video frames were downsampled to 128x128 pixels. Each 189-frame trial was treated as a single batch. We chose to label the moving mechanical equipement—two lick spouts and two levers—and the right paw (the left was always occluded). We hand labeled 50 frames and trained DLC [10] to obtain labels for the remaining frames. The lick spouts never move in the horizontal direction, so we only used their vertical position as

**Table 1. Number of neural dimensions returned by LocaNMF for each region/hemisphere in the two-view dataset.**

| Region Name | Acronym | Left | Right |
|---|---|---|---|
| Anterior Cingulate Area (dorsal) | ACAd | 8 | 7 |
| Auditory Area | AUD | 4 | 7 |
| Frontal Pole | FRP | 6 | 12 |
| Main Olfactory Bulb | MOB | 14 | 9 |
| Primary Motor Area | MOp | 7 | 6 |
| Secondary Motor Area | MOs | 14 | 14 |
| Pre-limbic Area | PL | 7 | 7 |
| Retrosplenial Area | RSP | 15 | 11 |
| Primary Somatosensory Area | SSp | 23 | 22 |
| Secondary Somatosensory Area | SSs | 7 | 7 |
| Temporal Association Area | TEa | 3 | 3 |
| Visual Area | VIS | 24 | 21 |

labels (for a total of two labels). The two levers always move together, and only along a specified path, so the combined movement is only one-dimensionsal; we therefore only used the $x$ coordinate of the left lever to fully represent the lever position (for a new total of three equipment-related labels). Finally, we use both the $x$ and $y$ coordinates of the paw label, for a total of five labels. We individually z-scored each of these five labels.

**Data splits.** We split data from each dataset into training (80%), validation (10%), and test trials (10%)—the first 8 trials are used for training, the next trial for validation, and the next trial for test. We then repeat this 10-block assignment of trials until no trials remain. Training trials are used to fit model parameters; validation trials are used for selecting hyperparameters and models; all plots and videos are produced using test trials, unless otherwise noted.

## PS-VAE: Model details

**Probabilistic formulation.** Here we detail the full probabilistic formulation of the PS-VAE, and provide a summary of the notation in Table 2. The PS-VAE transforms a frame $\mathbf{x}$ into a low-dimensional vector $\boldsymbol{\mu}(\mathbf{x}) = f(\mathbf{x})$ through the use of an encoder neural network $f(\cdot)$. Next we linearly transform this latent representation $\boldsymbol{\mu}(\mathbf{x})$ into the supervised and unsupervised latent representations. We define the supervised representation as

$$\mathbf{z}_s \sim \mathcal{N}\big(A\boldsymbol{\mu}(\mathbf{x}), \boldsymbol{\sigma}_s^2(\mathbf{x})\big). \tag{5}$$

The random variable $\mathbf{z}_s$ is normally distributed with a mean parameter defined by a linear mapping $A$ (which defines the supervised subspace), and a variance defined by another

**Table 2. A summary of the PS-VAE notation.**

| Symbol | Description |
|---|---|
| $\mathbf{x}$ | Video frame |
| $\mathbf{y}$ | Label vector |
| $\mathbf{z}_s$ | Supervised latent vector |
| $\mathbf{z}_u$ | Unsupervised latent vector |
| $\mathbf{z}_b$ | Background latent vector |
| $f(\cdot)$ | Convolutional encoder |
| $g(\cdot)$ | Convolutional (frame) decoder |
| $A$ | Matrix that defines supervised subspace |
| $B$ | Matrix that defines unsupervised subspace |
| $C$ | Matrix that defines background subspace |
| $D$ | Matrix that maps from supervised latents to labels |
| $\mathscr{L}_{\text{frames}}$ | Frame likelihood loss term |
| $\mathscr{L}_{\text{labels}}$ | Label likelihood loss term |
| $\mathscr{L}_{\text{KL}-s}$ | Kullback-Leibler (KL) loss term for supervised latents |
| $\mathscr{L}_{\text{ICMI}}$ | Index-code mutual information loss term for unsupervised latents |
| $\mathscr{L}_{\text{TC}}$ | Total Correlation loss term for unsupervised latents |
| $\mathscr{L}_{\text{DWKL}}$ | Dimension-wise KL loss term for unsupervised latents |
| $\mathscr{L}_{\text{orth}}$ | Orthogonalization loss term for full latent space |
| $\mathscr{L}_{\text{triplet}}$ | Triplet loss term for background latents |
| $\alpha$ | Weight on label reconstruction term |
| $\beta$ | Weight on Total Correlation term (unsupervised disentangling) |
| $\gamma$ | Weight on latent space orthogonalization term |
| $\delta$ | Weight on triplet loss term |

nonlinear transformation of the data $\boldsymbol{\sigma}_s^2(\cdot)$. This random variable contains the same number of dimensions as there are label coordinates, and each dimension is then required to reconstruct one of the label coordinates in $\mathbf{y}$ after application of another linear mapping

$$\mathbf{y} \sim \mathcal{N}(D\mathbf{z}_s + \mathbf{d}, \sigma_y I), \tag{6}$$

where $D$ is a diagonal matrix to allow for scaling of the $\mathbf{z}_s$'s.

Next we define the unsupervised representation as

$$\mathbf{z}_u \sim \mathcal{N}(B\boldsymbol{\mu}(\mathbf{x}), \boldsymbol{\sigma}_u^2(\mathbf{x})), \tag{7}$$

where the linear mapping $B$ defines the unsupervised subspace. We now construct the full latent representation $\mathbf{z} = [\mathbf{z}_s; \mathbf{z}_u]$ through concatenation and use $\mathbf{z}$ to reconstruct the observed video frames through the use of a decoder neural network $g(\cdot)$:

$$\mathbf{x} \sim \mathcal{N}(g(\mathbf{z}), \sigma_x I). \tag{8}$$

For simplicity we set $\sigma_y = 1$ and $\sigma_x = 1$.

We define the transformations $A$, $B$, and $D$ to be linear for several reasons. First of all, the linearity of $A$ and $B$ allows us to easily orthogonalize these subspaces, which we address more below. Second, the linearity (and additional diagonality) of $D$ ensures that it is invertible, simplifying the transformation from labels to latents that is useful for latent space traversals that we later use for qualitative evaluation of the models. Notably, these linear transformations all follow the nonlinear transformation $f(\cdot)$; as long as this is modeled with a high-capacity neural network, it should be able to capture the relevant nonlinear transformations and allow $A$, $B$, and $D$ to capture remaining linear transformations.

**Objective function.** We begin with a review of the standard ELBO objective for the VAE, then describe the modifications that result in the PS-VAE objective function.

**VAE objective**. The VAE [28, 29, 96] is a generative model composed of a likelihood $p_\theta(\mathbf{x}|\mathbf{z})$, which defines a distribution over the observed frames $\mathbf{x}$ conditioned on a set of latent variables $\mathbf{z}$, and a prior $p(\mathbf{z})$ over the latents. We define the distribution $q_\phi(\mathbf{z}|\mathbf{x})$ to be an approximation to the true posterior $p_\theta(\mathbf{z}|\mathbf{x})$. In the VAE framework this approximation uses a flexible neural network architecture to map the data to parameters of $q_\phi(\mathbf{z}|\mathbf{x})$. We define $\boldsymbol{\mu}_\phi(\mathbf{x}) = f_\phi(\mathbf{x})$ to be the deterministic mapping of the data $\mathbf{x}$ through an arbitrary neural network $f_\phi(\cdot)$, resulting in a deterministic latent space. In the VAE framework $\boldsymbol{\mu}_\phi$ can represent the natural parameters of an exponential family distribution, such as the mean in a Gaussian distribution. Framed in this way, inference of the (approximate) posterior is now recast as an optimization problem which finds values of the parameters $\phi$ and $\theta$ that minimize the distance (KL divergence) between the true and approximate posterior.

Unfortunately, we cannot directly minimize the KL divergence between $p_\theta(\mathbf{z}|\mathbf{x})$ and $q_\phi(\mathbf{z}|\mathbf{x})$ because $p_\theta(\mathbf{z}|\mathbf{x})$ is the unknown distribution that we want to find in the first place. Instead we maximize the Evidence Lower Bound (ELBO), defined as

$$\begin{aligned} \mathcal{L}'_{\mathrm{ELBO}}(\theta, \phi) &= \log p_\theta(\mathbf{x}) - \mathrm{KL}[q_\phi(\mathbf{z}|\mathbf{x})||p_\theta(\mathbf{z}|\mathbf{x})] \\ &= \mathbb{E}_{q_\phi(\mathbf{z}|\mathbf{x})}[\log p_\theta(\mathbf{x}|\mathbf{z})] - \mathrm{KL}[q_\phi(\mathbf{z}|\mathbf{x})||p(\mathbf{z})]. \end{aligned}$$

In practice we have a finite dataset $\{\mathbf{x}_n\}_{n=1}^N$ and optimize

$$\mathcal{L}_{\mathrm{ELBO}}(\theta, \phi) = \frac{1}{N} \sum_{n=1}^N \mathbb{E}_{q_\phi(\mathbf{z}_n|\mathbf{x}_n)}[\log p_\theta(\mathbf{x}_n|\mathbf{z}_n)] - \mathrm{KL}[q_\phi(\mathbf{z}_n|\mathbf{x}_n)||p(\mathbf{z}_n)].$$

To simplify notation, we follow [97] and define the approximate posterior as $q(\mathbf{z}|n) \triangleq q_\phi(\mathbf{z}_n|\mathbf{x}_n)$,

drop other subscripts, and treat $n$ as a random variable with a uniform distribution $p(n)$. With these notational changes we can rewrite the ELBO as

$$\mathscr{L}_{\text{ELBO}} = \mathbb{E}_{p(n)}[\mathbb{E}_{q(\mathbf{z}|n)}[\log p(\mathbf{x}|\mathbf{z})]] - \mathbb{E}_{p(n)}[\text{KL}[q(\mathbf{z}|n)||p(\mathbf{z})]], \tag{9}$$

and define $\mathscr{L}_{\text{frames}} \triangleq \mathbb{E}_{p(n)}[\mathbb{E}_{q(\mathbf{z}|n)}[\log p(\mathbf{x}|\mathbf{z})]]$. This objective function can be easily optimized when the latents are modeled as a continuous distribution using the reparameterization trick with stochastic gradient descent [28].

**PS-VAE objective**. To model the labels $\mathbf{y}$, we replace the VAE's input likelihood $p(\mathbf{x}|\mathbf{z})$ by a joint likelihood of inputs and labels $p(\mathbf{x}, \mathbf{y}|\mathbf{z})$. We make the simplifying assumption that the frames $\mathbf{x}$ and the labels $\mathbf{y}$ are conditionally independent given the latent $\mathbf{z}$ [30, 31]: $p(\mathbf{x}, \mathbf{y}|\mathbf{z}) = p(\mathbf{x}|\mathbf{z})p(\mathbf{y}|\mathbf{z})$, so that the log-likelihood term splits in two:

$$\begin{aligned}
\mathbb{E}_{p(n)}[\mathbb{E}_{q(\mathbf{z}|n)}[\log p(\mathbf{x}, \mathbf{y}|\mathbf{z})]] &= \mathbb{E}_{p(n)}[\mathbb{E}_{q(\mathbf{z}|n)}[\log p(\mathbf{x}|\mathbf{z})p(\mathbf{y}|\mathbf{z})]] \\
&= \mathbb{E}_{p(n)}[\mathbb{E}_{q(\mathbf{z}|n)}[\log p(\mathbf{x}|\mathbf{z})]] + \mathbb{E}_{p(n)}[\mathbb{E}_{q(\mathbf{z}|n)}[\log p(\mathbf{y}|\mathbf{z})]] \\
&\triangleq \mathscr{L}_{\text{frames}} + \mathscr{L}_{\text{labels}}.
\end{aligned} \tag{10}$$

Next we turn to the KL term of the VAE objective function (second term in Eq 9). We assume a fully factorized prior of the form $p(\mathbf{z}) = \prod_i p(z_i)$ (as well as a variational distribution that is factorized into separate supervised and unsupervised distributions), which again simplifies the objective function by allowing us to split this term between the supervised and unsupervised latents:

$$\begin{aligned}
\mathbb{E}_{p(n)}[\text{KL}[q(\mathbf{z}|n)||p(\mathbf{z})]] &= \mathbb{E}_{p(n)}[\text{KL}[q(\mathbf{z}_s, \mathbf{z}_u|n)||p(\mathbf{z}_s, \mathbf{z}_u)]] \\
&= \mathbb{E}_{p(n)}[\text{KL}[q(\mathbf{z}_s|n)q(\mathbf{z}_u|n)||p(\mathbf{z}_s)p(\mathbf{z}_u)]] \\
&= \mathbb{E}_{p(n)}[\text{KL}[q(\mathbf{z}_s|n)||p(\mathbf{z}_s)]] + \mathbb{E}_{p(n)}[\text{KL}[q(\mathbf{z}_u|n)||p(\mathbf{z}_u)]] \\
&\triangleq \mathscr{L}_{\text{KL}-s} + \mathscr{L}_{\text{KL}-u}.
\end{aligned} \tag{11}$$

$\mathscr{L}_{\text{KL}-s}$, the KL term for $\mathbf{z}_s$, will remain unmodified, as the labels will be responsible for structuring this part of the representation. To enforce a notion of "disentangling" on the unsupervised latents we adopt the KL decomposition proposed in [42, 45]:

$$\begin{aligned}
\mathscr{L}_{\text{KL}-u} &= \text{KL}[q(\mathbf{z}_u, n)||q(\mathbf{z}_u)p(n)] + \text{KL}\left[q(\mathbf{z}_u)||\prod_j q(z_{u,j})\right] + \sum_j \text{KL}[q(z_{u,j})||p(z_{u,j})] \\
&\triangleq \mathscr{L}_{\text{ICMI}} + \mathscr{L}_{\text{TC}} + \mathscr{L}_{\text{DWKL}},
\end{aligned} \tag{12}$$

where $z_{u,j}$ denotes the $j$th dimension of $\mathbf{z}_u$. The first term $\mathscr{L}_{\text{ICMI}}$ is the index-code mutual information [97], which measures the mutual information between the data and the latent variable; generally we do not want to penalize this term too aggressively, since we want to maintain the relationship between the data and its corresponding latent representation. Nevertheless, slight penalization of this term does not seem to hurt, and may even help in some cases [45]. The second term $\mathscr{L}_{\text{TC}}$ is the Total Correlation (TC), one of many generalizations of mutual information to more than two random variables. The TC has been the focus many recent papers on disentangling [42–45], as penalizing this term forces the model to find statistically independent latent dimensions. Therefore we add a hyperparameter $\beta$ that allows us to control the strength of this penalty. The final term $\mathscr{L}_{\text{DWKL}}$ is the dimension-wise KL, which measures the distance between the approximate posterior and the prior for each dimension individually.

We also add another hyperparameter $\alpha$ to the log-likelihood of the labels, so that we can tune the extent to which this information shapes the supervised subspace. Finally, we add a term (with its own hyperparameter $\gamma$) that encourages the subspaces defined by the matrices $A$

and $B$ to be orthogonal, $\mathscr{L}_{\text{orth}} = ||UU^T - I||$, where $U = [A; B]$. The final objective function is given by

$$\mathscr{L}_{\text{PS-VAE}} = \mathscr{L}_{\text{frames}} + \alpha\mathscr{L}_{\text{labels}} - \mathscr{L}_{\text{KL-s}} - \mathscr{L}_{\text{ICMI}} - \beta\mathscr{L}_{\text{TC}} - \mathscr{L}_{\text{DWKL}} - \gamma\mathscr{L}_{\text{orth}}. \tag{13}$$

This objective function is no longer strictly a lower bound on the log probability due to the addition of $\alpha$, which allows $\mathscr{L}_{\text{PS-VAE}}$ to be greater than $\mathscr{L}_{\text{ELBO}}$. Nevertheless, we find this objective function produces good results. See the following section for additional details on computing the individual terms of this objective function. We discuss the selection of the hyperparameters $\alpha$, $\beta$, and $\gamma$ below.

**Computing the PS-VAE objective function.** The frame log-likelihood ($\mathscr{L}_{\text{frames}}$), label log-likelihood ($\mathscr{L}_{\text{labels}}$), KL divergence for the supervised subspace ($\mathscr{L}_{\text{KL-s}}$), and orthogonality constraint ($\mathscr{L}_{\text{orth}}$) in Eq 13 are all standard computations. The remaining terms in the objective function cannot be computed exactly when using stochastic gradient updates because $q(\mathbf{z}) = \sum_{n=1}^{N} q(\mathbf{z}|n)p(n)$ requires iterating over the entire dataset. [45] introduced the following Monte Carlo approximation from a minibatch of samples $\{n_1, \ldots, \ldots n_M\}$:

$$\mathbb{E}_{q(\mathbf{z})}[\log q(\mathbf{z})] \approx \frac{1}{M}\sum_{i=1}^{M}\left[\log \frac{1}{NM}\sum_{j=1}^{M} q(\mathbf{z}(n_i)|n_j)\right], \tag{14}$$

which allows for the batch-wise estimation of the remaining terms. The crucial quantity $q(\mathbf{z}(n_i)|n_j)$ is computed by evaluating the probability of observation $i$ under the posterior of observation $j$. A full implementation of these approximations can be found in the accompanying PS-VAE code repository.

**Index-code mutual information**. We first look at the term $\mathscr{L}_{\text{ICMI}}$ in Eq 13. In what follows we drop the $u$ subscript from $\mathbf{z}_u$ for clarity.

$$
\begin{aligned}
\text{KL}[q(\mathbf{z}, n)||q(\mathbf{z})p(n)] &= \mathbb{E}_{q(\mathbf{z},n)}[\log q(\mathbf{z}, n) - \log q(\mathbf{z})p(n)] \\
&= \mathbb{E}_{q(\mathbf{z},n)}[\log q(\mathbf{z}, n) - \log q(\mathbf{z}) - \log p(n)]
\end{aligned}
$$

Let's look at the first two expectations individually.

First,

$$
\begin{aligned}
\mathbb{E}_{q(\mathbf{z},n)}[\log q(\mathbf{z}, n)] &= \mathbb{E}_{q(\mathbf{z},n)}[\log q(\mathbf{z}|n)p(n)] \\
&= \mathbb{E}_{p(n)}\mathbb{E}_{q(\mathbf{z}|n)}[\log q(\mathbf{z}|n)] + \mathbb{E}_{q(\mathbf{z},n)}[\log p(n)] \\
&\approx \frac{1}{M}\sum_{i=1}^{M}\log q(\mathbf{z}(n_i)|n_i) + \mathbb{E}_{q(\mathbf{z},n)}[\log p(n)].
\end{aligned}
$$

Next,

$$
\begin{aligned}
\mathbb{E}_{q(\mathbf{z},n)}[\log q(\mathbf{z})] &= \int_z \sum_n q(\mathbf{z}, n)\log q(\mathbf{z}) \\
&= \int_z q(\mathbf{z})\log q(\mathbf{z}) \\
&= \mathbb{E}_{q(\mathbf{z})}[\log q(\mathbf{z})] \\
&\approx \frac{1}{M}\sum_{i=1}^{M}\left[\log \frac{1}{NM}\sum_{j=1}^{M} q(\mathbf{z}(n_i)|n_j)\right]
\end{aligned}
$$

Putting it all together,

$$
\begin{aligned}
\mathrm{KL}(q(\mathbf{z},n)||q(\mathbf{z})p(n)) \quad &\approx \quad \frac{1}{M}\sum_{i=1}^{M}\log q(\mathbf{z}(n_i)|n_i) - \frac{1}{M}\sum_{i=1}^{M}\left[\log\frac{1}{NM}\sum_{j=1}^{M}q(\mathbf{z}(n_i)|n_j)\right] \\
&= \quad \frac{1}{M}\sum_{i=1}^{M}\left[\log q(\mathbf{z}(n_i)|n_i) - \log\sum_{j=1}^{M}q(\mathbf{z}(n_i)|n_j) + \log NM\right]
\end{aligned}
$$

**Total Correlation**. We next look at the term $\mathscr{L}_{\mathrm{TC}}$ in Eq 13; in what follows $\mathbf{z}_l$ denotes the $l^{\mathrm{th}}$ dimension of the vector $\mathbf{z}$.

$$
\begin{aligned}
\mathrm{KL}\left[q(\mathbf{z})||\prod_l q(\mathbf{z}_l)\right] \quad &= \quad \mathbb{E}_{q(\mathbf{z})}\left[\log q(\mathbf{z}) - \sum_l q(\mathbf{z}_l)\right] \\
&\approx \quad \frac{1}{M}\sum_{i=1}^{M}\left[\log\frac{1}{NM}\sum_{j=1}^{M}q(\mathbf{z}(n_i)|n_j) - \sum_l\log\frac{1}{NM}\sum_{j=1}^{M}q(\mathbf{z}(n_i)_l|n_j)\right] \\
&= \quad \frac{1}{M}\sum_{i=1}^{M}\left[\log\sum_{j=1}^{M}q(\mathbf{z}(n_i)|n_j) - \sum_l\log\sum_{j=1}^{M}q(\mathbf{z}(n_i)_l|n_j)\right. \\
&\qquad + \quad (L-1)\log NM\Bigg]
\end{aligned}
$$

**Dimension-wise KL**. Finally, we look at the term $\mathscr{L}_{\mathrm{DWKL}}$ in Eq 13.

$$
\begin{aligned}
\sum_l\mathrm{KL}[q(\mathbf{z}_l)||p(\mathbf{z}_l)] \quad &= \quad \sum_l\mathbb{E}_{q(\mathbf{z}_l)}[\log q(\mathbf{z}_l) - \log p(\mathbf{z}_l)] \\
&= \quad \mathbb{E}_{q(\mathbf{z})}\sum_l[\log q(\mathbf{z}_l) - \log p(\mathbf{z}_l)] \\
&\approx \quad \frac{1}{M}\sum_{i=1}^{M}\sum_l\left[\log\frac{1}{NM}\sum_{j=1}^{M}q(\mathbf{z}(n_i)_l|n_j) - \log p(\mathbf{z}(n_i)_l)\right] \\
&= \quad \frac{1}{M}\sum_{i=1}^{M}\sum_l\left[\log\sum_{j=1}^{M}q(\mathbf{z}(n_i)_l|n_j) - \log p(\mathbf{z}(n_i)_l) - \log NM\right]
\end{aligned}
$$

where the second equality assumes that $q(\mathbf{z})$ is a factorized approximate posterior.

**Training procedure.** We trained all models using the Adam optimizer [98] for 200 epochs with a learning rate of $10^{-4}$ and no regularization, which we found to work well across all data-sets. Batch sizes were dataset-dependent, ranging from 100 frames to 189 frames. All KL terms and their decompositions were annealed for 100 epochs, which we found to help with latent collapse [99]. For example, the weight on the KL term of the VAE was linearly increased from 0 to 1 over 100 epochs. For the PS-VAE, the weights on $\mathscr{L}_{\mathrm{ICMI}}$ and $\mathscr{L}_{\mathrm{DWKL}}$ were increased from 0 to 1, while the weight on $\mathscr{L}_{\mathrm{TC}}$ was increased from 0 to $\beta$.

**Model architecture.** For all models (VAE, $\beta$-TC-VAE, PS-VAE, MSPS-VAE) we used a similar convolutional architecture; details differ in how the latent space is defined. See Table 3 for network architecture details of the standard VAE.

## PS-VAE: Hyperparameter selection

The PS-VAE objective function (Eq 13) is comprised of terms for the reconstruction of the video frames ($\mathscr{L}_{\mathrm{frames}}$), reconstruction of the labels ($\mathscr{L}_{\mathrm{labels}}$, controlled by $\alpha$), factorization of

**Table 3. Convolutional VAE architecture for the IBL dataset using *N* latents.** Kernel size and stride size are defined as (x pixels, y pixels); padding size is defined as (left, right, top, bottom); output size is defined as (x pixels, y pixels, channels). Reparameterization details not included.

| Layer | Type | Channels | Kernel Size | Stride Size | Zero Padding | Output Size |
|-------|------|----------|-------------|-------------|--------------|-------------|
| 0 | conv | 32 | (5, 5) | (2, 2) | (1, 2, 1, 2) | (96, 96, 32) |
| 1 | conv | 64 | (5, 5) | (2, 2) | (1, 2, 1, 2) | (48, 48, 64) |
| 2 | conv | 128 | (5, 5) | (2, 2) | (1, 2, 1, 2) | (24, 24, 128) |
| 3 | conv | 256 | (5, 5) | (2, 2) | (1, 2, 1, 2) | (12, 12, 256) |
| 4 | conv | 512 | (5, 5) | (2, 2) | (1, 2, 1, 2) | (6, 6, 512) |
| 5 | dense | *N* | NA | NA | NA | (1, 1, *N*) |
| 6 | dense | 36 | NA | NA | NA | (1, 1, 36) |
| 7 | reshape | NA | NA | NA | NA | (6, 6, 1) |
| 8 | conv transpose | 256 | (5, 5) | (2, 2) | (1, 2, 1, 2) | (12, 12, 256) |
| 9 | conv transpose | 128 | (5, 5) | (2, 2) | (1, 2, 1, 2) | (24, 24, 128) |
| 10 | conv transpose | 64 | (5, 5) | (2, 2) | (1, 2, 1, 2) | (48, 48, 64) |
| 11 | conv transpose | 32 | (5, 5) | (2, 2) | (1, 2, 1, 2) | (96, 96, 32) |
| 12 | conv transpose | 1 | (5, 5) | (2, 2) | (1, 2, 1, 2) | (192, 192, 1) |

the unsupervised latent space ($\mathscr{L}_{TC}$, controlled by $\beta$), and orthogonality of the entire latent space ($\mathscr{L}_{orth}$, controlled by $\gamma$). We explore these terms one at a time with the head-fixed mouse dataset and highlight the sensitivity of the associated hyperparameters; the identical analysis for the remaining datasets can be found in S10 Fig (freely moving mouse), S11 Fig (mouse face), and S12 Fig (two-view).

The hyperparameter $\alpha$ controls the strength of the label log-likelihood term, which needs to be balanced against the frame log-likelihood term. To investigate the effect of $\alpha$ we set the default values of $\beta = 1$ and $\gamma = 0$. Increasing $\alpha$ leads to better label reconstructions across a range of latent dimensionalities, at the cost of worse frame reconstructions (Fig 11A and 11B). However, the increase in frame reconstruction error is quite small, and robust to $\alpha$ over several orders of magnitude. Recall that we first z-scored each label individually, which affects the magnitude of $\alpha$. By performing this z-scoring for all datasets, we find similar results across the same range of $\alpha$ values (S10–S12 Figs). We find that $\alpha = 1000$ is a reasonable default value for this hyperparameter, as it provides a good trade-off between frame and label reconstruction quality.

We next explore the remaining hyperparameters $\beta$ and $\gamma$. To do so we choose a 6D model, which contains a 4D supervised subspace and a 2D unsupervised subspace. This choice admits easy visualization of the unsupervised subspace, and is the choice we made for the main text. We first show that $\beta$ and $\gamma$ have little to no effect on either the frame reconstruction (Fig 11C) or the label reconstruction (Fig 11D). This allows us to freely choose these hyperparameters without worrying about their effect on the reconstruction terms. Next we look at the effect of $\beta$ and $\gamma$ on the three terms of the KL decomposition for the unsupervised subspace (Eq 12). The first term, the index-code mutual information, decreases as a function of $\beta$, even though it is not directly penalized by $\beta$ (Fig 11E). This decrease is in general undesirable, since it indicates that the latent representation contains less information about the corresponding data point. The second term, the Total Correlation (TC), also decreases as a function of $\beta$, as desired (Fig 11F). Finally, the dimension-wise KL term also changes as a function of $\beta$, even though it is not directly penalized (Fig 11G). The increase when $\gamma = 1000$ is in general undesirable, since it indicates the aggregate posterior is becoming less like the prior. To conclude, as we continue to increase the value of $\beta$ we will continue to see a decrease in the TC, but these curves demonstrate that a small TC can be accompanied by other undesirable features of the latent

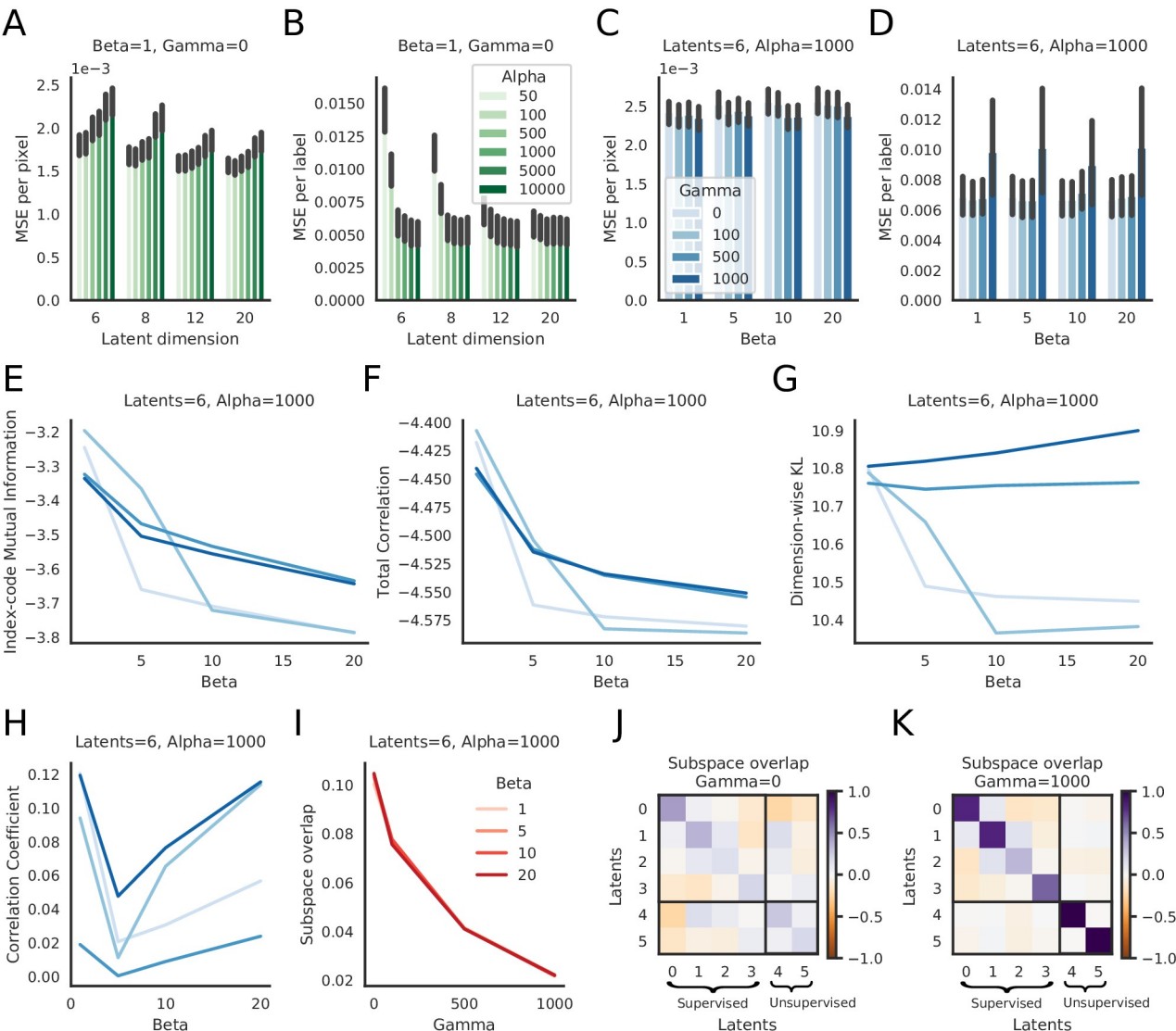

**Fig 11. PS-VAE hyperparameter selection for the head-fixed mouse dataset. A**: MSE per pixel as a function of latent dimensionality and the hyperparameter $\alpha$, which controls the strength of the label reconstruction term. The frame reconstruction is robust across many orders of magnitude. **B**: MSE per label as a function of latent dimensionality and $\alpha$. As the latent dimensionality increases the model becomes more robust to $\alpha$, but is sensitive to this value when the model has few latents due to the strong tradeoff between frame and label reconstruction. Subsequent panels detail $\beta$ and $\gamma$ with a 6D model and $\alpha = 1000$. **C**: MSE per pixel as a function of $\beta$ and $\gamma$; frame reconstruction is robust to both of these hyperprameters. **D**: MSE per label as a function of $\beta$ and $\gamma$; label reconstruction is robust to both of these hyperprameters. **E**: Index code mutual information (ICMI; see Eq 13) as a function of $\beta$ and $\gamma$. The ICMI, although not explicitly penalized by $\beta$, is affected by this hyperparameter. **F**: Total Correlation (TC) as a function of $\beta$ and $\gamma$. Increasing $\beta$ decreases the TC as desired. **G**: Dimension-wise KL (DWKL) as a function of $\beta$ and $\gamma$. The DWKL, although not explicitly penalized by $\beta$, is affected by this hyperparameter. **H**: Pearson correlation in the model's 2D unsupervised subspace as a function of $\beta$ and $\gamma$. **I**: The subspace overlap as defined by $\|UU^T - I\|^2$ (where $U = [A;B]$ and $I$ the identity) as a function of $\beta$ and $\gamma$. Increasing $\gamma$ leads to an orthogonalized latent space, while varying $\beta$ has no effect. **J**: Example subspace overlap matrix ($UU^T$) for $\gamma = 0$. The upper left 4x4 block represents the supervised subspace, the lower right 2x2 block represents the unsusupervised subspace. **K**: Example subspace overlap matrix for $\gamma = 1000$; the subspace is close to orthogonal. Error bars in panels A-D represent 95% bootstrapped confidence interval over test trials; line plots in panels E-H are the mean values over test trials, and confidence intervals are omitted for clarity.

representation. Therefore we cannot simply choose the model with the lowest TC value as the one that is most "interpretable."

As an alternative, simple measure of interpretability, we compute the Pearson correlation coefficient between each pair of latent dimensions. The motivation for this measure is

that it quantifies the (linear) statistical relationship between each pair of dimensions; while not as general as the TC term, we are able to compute it exactly over each trial, and find empirically that it is a good indicator of interpretability. We find correlations decrease and then increase for increasing values of $\beta$ (Fig 11H). The subsequent increase is due to the trade off in the objective function between the Total Correlation and the other KL terms as described above; we find a balance is struck with regards to the Pearson correlation at $\beta = 5$.

Increasing the hyperparameter $\gamma$ forces the entire latent space (supervised and unsupervised) to become more orthogonal (Fig 11I), which may aid in interpretability since each dimension can be independently manipulated [37]. Fig 11J and 11K show examples of the subspace overlap matrix $UU^T$, where $U = [A; B]$ is the concatenation of the mapping into the supervised subspace ($A$) with the mapping into the unsupervised subspace ($B$). At $\gamma = 1000$ the subspace is close to orthogonal.

Given these observations, we conclude that in a 6D model, setting the hyperparameters to $\alpha = 1000$, $\beta = 5$, $\gamma = 500$ should provide the most interpretable representation on the head-fixed mouse dataset. Indeed, we find that this model does provide a good representation (Fig 2), although we note that other combinations of $\beta$ and $\gamma$ can provide good qualitative representations as well. We repeated this analysis on the freely moving mouse (S10 Fig), mouse face (S11 Fig), and two-video datasets (S12 Fig), and using the same criteria as above chose models with the hyperparameters listed in S1 Table.

We distill these steps into a general hyperparameter selection process. We found it helpful to start this process using a 2D unsupervised subspace, for ease of visualization; if more unsupervised latents are desired this process can be repeated for three or four unsupervised latents. In the datasets considered in this paper we found the PS-VAE typically did not utilize more than 3 or 4 unsupervised latents (see Qualitative model comparisons in the Results), a phenomenon referred to as "latent collapse" [99].

**PS-VAE hyperparameter selection process**:

**Step 0**: Individually z-score labels before model fitting.

**Step 1**: Set the dimensionality of the unsupervised subspace to 2.

**Step 2**: Set $\beta = 1$, $\gamma = 0$, and fit models for $\alpha = [50, 100, 500, 1000, 5000]$. Choose the value of $\alpha$ that provides a desirable trade-off between frame reconstruction and label reconstruction (call this $\alpha'$).

**Step 3**: Set $\alpha = \alpha'$ and fit models for all combinations of $\beta = [1, 5, 10, 20]$ and $\gamma = [100, 500, 1000]$. Choose the $\alpha$, $\beta$ combination with the lowest correlation coefficient averaged over all pairs of unsupervised dimensions (as in Fig 11H) (call these $\beta'$ and $\gamma'$).

**Step 4** [optional]: Set $\alpha = \alpha'$, $\beta = \beta'$, $\gamma = \gamma'$ and refit the PS-VAE using several random weight initializations, which may result in qualitatively and/or quantitatively improved models (using latent traversals and correlation coefficients, respectively).

**Step 5** [optional]: Increase the dimensionality of the unsupervised subspace by 1, then repeat Steps 2–4.

This process requires fitting 17 models for a single dimensionality of the unsupervised subspace: 5 models for Step 2 and 12 models for Step 3. This process can take several days of GPU time, depending on available hardware and the size of the dataset (we were able to fit single models in approximately 4 hours using an Nvidia GeForce GTX 1080 graphics card). Streamlining this hyperparameter selection process is a focus of future work.

## PS-VAE: Latent traversals

The generation of new behavioral video frames is a useful technique for investigating the latent representation learned by a model. We employ this technique in the figures (e.g. Fig 2) and to greater effect in the videos (S15 Fig). To do so, we isolate a single dimension (supervised or unsupervised), and create a series of frames as we move along that dimension in the latent space. Note that the resulting frames are fully generated by the model; we are not selecting real frames from the dataset. When producing these "latent traversals" we typically range from the $10^{th}$ to the $90^{th}$ percentile value of the chosen dimension, computed across the latent representations of the training data.

We first choose a frame $\mathbf{x}$ and push it through the encoder $f(\cdot)$ to produce the latent vector $\boldsymbol{\mu}(\mathbf{x}) = f(\mathbf{x})$, which is used to compute the posterior:

$$\mathbf{z}_s \sim \mathcal{N}(A\boldsymbol{\mu}(\mathbf{x}), \boldsymbol{\sigma}_s^2(\mathbf{x}))$$
$$\mathbf{z}_u \sim \mathcal{N}(B\boldsymbol{\mu}(\mathbf{x}), \boldsymbol{\sigma}_u^2(\mathbf{x})).$$

For this procedure we do not sample from the posterior but rather use the posterior means, so that $\hat{\mathbf{z}}_s = A\boldsymbol{\mu}(\mathbf{x})$ and $\hat{\mathbf{z}}_u = B\boldsymbol{\mu}(\mathbf{x})$.

In order to generate frames through manipulation of the supervised latent representation, $\hat{\mathbf{z}}_s$ is first converted to an estimate of the labels $\hat{\mathbf{y}}$ through a linear, invertible transform:

$$\hat{\mathbf{y}} = D\hat{\mathbf{z}}_s + \mathbf{d},$$

where $D$ is a diagonal matrix and we use the notation $D^{-1}$ to denote the matrix with inverted values on the diagonal. We can now choose an arbitrary set of target values for the $x$, $y$ coordinates of a specific label (e.g. left paw), and fix the values of all other labels that accompany the frame $\mathbf{x}$. We denote this manipulated label vector as $\bar{\mathbf{y}}$. After forming $\bar{\mathbf{y}}$ we transform this vector into the latent representation used by the frame decoder:

$$\bar{\mathbf{z}}_s = D^{-1}(\bar{\mathbf{y}} - \mathbf{d})$$

and form the new latent vector $\bar{\mathbf{z}} = [\bar{\mathbf{z}}_s; \hat{\mathbf{z}}_u]$ (without additional sampling), and generate a new frame as $\bar{\mathbf{x}} = g_\theta(\bar{\mathbf{z}})$.

Note that we could also directly manipulate the supervised representation $\hat{\mathbf{z}}_s$, rather than the transformed representation $\hat{\mathbf{y}}$. We choose to do that latter, since the manipulated values in pixel space are easier for a human to understand—for example, we can think about shifting the horizontal position of a paw by a certain number of pixels. Regardless of whether the traversal is performed in the pixel space or the latent space, the results will be the same due to the invertibility of $D$.

In order to generate frames through manipulation of the unsupervised latent representation, we change one or more values of $\hat{\mathbf{z}}_u$ (denoted as $\bar{\mathbf{z}}_u$) while keeping all values of $\hat{\mathbf{z}}_s$ fixed. We then form the new latent vector $\bar{\mathbf{z}} = [\hat{\mathbf{z}}_s; \bar{\mathbf{z}}_u]$ (without additional sampling), and generate a new frame as $\bar{\mathbf{x}} = g_\theta(\bar{\mathbf{z}})$.

## MSPS-VAE: Model details

The PS-VAE formulation introduced above does not adequately address inter-session variability when the model is trained on videos from multiple experimental sessions (Fig 10C). This variability can come from many sources, including subjects with different physical features, inconsistent camera placements that produce different angles or fields of view, and inconsistent lighting. Here we introduce a natural extension to the PS-VAE, the multi-session PS-VAE (MSPS-VAE), intended to handle this inter-session variability (Fig 12).

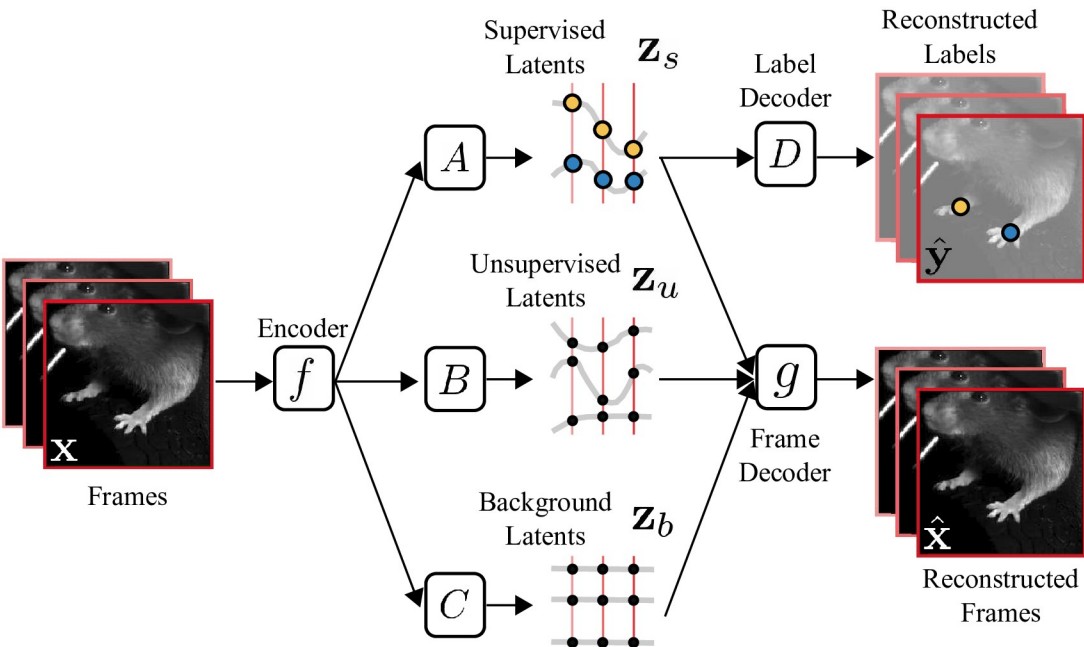

**Fig 12. Overview of the multi-session Partitioned Subspace VAE (MSPS-VAE).** The MSPS-VAE finds a low-dimensional latent representation of a behavioral video that is partitioned into three subspaces: one subspace contains the *supervised* latent variables $\mathbf{z}_s$ (present in the PS-VAE), a second subspace contains the *unsupervised* latent variables $\mathbf{z}_u$ (present in the PS-VAE), and the third subspace contains the *background* latent variables $\mathbf{z}_b$, which capture inter-session variability, and are new to the MSPS-VAE formulation. The supervised latent variables are required to reconstruct user-supplied labels, while all three sets of latent variables are together required to reconstruct the video frames.

Recall that the PS-VAE transforms a frame $\mathbf{x}$ into a low-dimensional vector $\boldsymbol{\mu}(\mathbf{x}) = f(\mathbf{x})$ through the use of an encoder neural network $f(\cdot)$. This low-dimensional vector is then linearly transformed into the supervised latents $\mathbf{z}_s$ and the unsupervised latents $\mathbf{z}_u$ through the mappings $A$ and $B$, respectively. We now add a third "background" subspace $\mathbf{z}_b = C\,\boldsymbol{\mu}(\mathbf{x})$ (this representation is not probabilistic), and increase the dimensionality of $\boldsymbol{\mu}$ such that $|\boldsymbol{\mu}| = |\mathbf{z}_s| + |\mathbf{z}_u| + |\mathbf{z}_b|$. Our goal is to encourage $\mathbf{z}_b$ to capture static differences between sessions, leaving the other subspaces to capture dynamic behaviors. In order to do so we employ a triplet similarity loss [54], which optimizes the background latents $\mathbf{z}_b$ such that data points from the same session are closer to each other than those from different sessions (more details below). We then construct the full latent representation $\mathbf{z} = [\mathbf{z}_s; \mathbf{z}_u; \mathbf{z}_b]$ through concatenation and, as before, use $\mathbf{z}$ to reconstruct the observed video frame with a convolutional decoder neural network (Eq 4).

We now describe the triplet similarity loss in more detail, and how it is incorporated into the MSPS-VAE objective function. In the triplet loss terminology, there are "anchor" points $a_i$, "positive" points $p_i$, and "negative" points $n_i$. The triplet loss is then defined as

$$L_t(a_i, p_i, n_i) = \max\{d(a_i, p_i) - d(a_i, n_i) + m, 0\} \tag{15}$$

where $d(\cdot, \cdot)$ is any distance function (we use Euclidean distance for simplicity), and $m$ is a margin typically set to 1. This loss pulls the positive point towards the anchor point, and pushes the negative point away from the anchor point.

Concretely, we load a batch of data $\mathbf{X}_k$ from session $k$ and a batch $\mathbf{X}_j$ from session $j$ and split each into three equally sized groups, denoted for example by $\mathbf{X}_k = \{\mathbf{X}_k^1, \mathbf{X}_k^2, \mathbf{X}_k^3\}$. Then an

aggregate triplet loss can be simply computed as

$$\mathscr{L}_{\text{triplet}} = \sum_i L_t(\mathbf{X}_{k,i}^1, \mathbf{X}_{k,i}^2, \mathbf{X}_{j,i}^3) + \sum_i L_t(\mathbf{X}_{j,i}^1, \mathbf{X}_{j,i}^2, \mathbf{X}_{k,i}^3),$$ (16)

where $\mathbf{X}_{m,i}^n$ denotes the $i^{th}$ data point of group $\mathbf{X}_m^n$ (i.e. $L_t$ is computed for each triplet of data points individually then summed across the batch). We add another hyperparameter $\delta$ to regulate the influence of the triplet loss on the overall cost function.

With the addition of the new subspace we introduce a new linear transformation $C$; we would like the subspace defined by $C$ to be orthogonal to the other subspaces defined by $A$ and $B$, which before were encouraged to be orthogonal with $\mathscr{L}_{\text{orth}}$. Because we have also introduced a new hyperparameter with the triplet loss, we modify the way in which we handle $\mathscr{L}_{\text{orth}}$ to keep the hyperparameter search space of manageable size. Whereas before we merely encouraged orthogonality among $A$ and $B$, we will now *enforce* orthogonality among $A$, $B$, and $C$ by explicitly defining these matrices to be orthogonal, and leave them fixed throughout the optimization procedure. Because these linear transformations follow a high-capacity convolutional neural network $f(\cdot)$, we can force $f(\cdot)$ to produce the necessary nonlinear transformation such that $\{A, B, C\}$ produce good subspaces that minimize their respective losses. Therefore, the final MSPS-VAE objective function is given by

$$\mathscr{L}_{\text{MSPS-VAE}} = \mathscr{L}_{\text{frames}} + \alpha\mathscr{L}_{\text{labels}} - \mathscr{L}_{\text{KL-s}} - \mathscr{L}_{\text{ICMI}} - \beta\mathscr{L}_{\text{TC}} - \mathscr{L}_{\text{DWKL}} - \delta\mathscr{L}_{\text{triplet}}.$$ (17)

The training procedure remains as described above, except we load batches from two different sessions on each iteration to compute $\mathscr{L}_{\text{triplet}}$. The other terms are computed and summed across both batches. The model architecture also remains as described in above.

Finally, we note that the introduction of $\mathbf{z}_b$ also requires a choice for the dimensionality of this vector; we chose a small number, $|\mathbf{z}_b| = 3$, but found that other choices ($|\mathbf{z}_b| = 2, 4$) did not noticeably affect performance.

## MSPS-VAE: Hyperparameter selection

The hyperparameter selection process for the MSPS-VAE follows that of the PS-VAE, where we replace a search over $\gamma$ (controlling the orthogonality of the subspaces) with a search over $\delta$ (controlling the triplet loss). S13 Fig shows the results of this hyperparameter search. We note that $\delta$ has little to no effect on either the frame reconstruction (S13C Fig), the label reconstruction (S13D Fig), or the terms from the KL decomposition (S13E–S13G Fig). In this dataset there is some effect on the correlation coefficient (S13H Fig), the metric previously used with the PS-VAE to choose $\beta$ and $\gamma$. We introduce another metric here to assist in the selection of $\beta$ and $\delta$: the accuracy of a linear classifier trained to predict session identity from the unsupervised representation. This metric should be near chance (1 / number of sessions) for models that have successfully learned to place inter-session variability in the background subspace. This metric is heavily influenced by the value of $\delta$, and thus provides a clearer distinction between models (S13I Fig). Finally, we also look at the triplet loss as a function of $\beta$ and $\delta$ and see that $\beta$ does not have a large influence on these values (S13J Fig). For clarity and completeness we provide the full search process below.

**MSPS-VAE hyperparameter selection process**:

**Step 0**: Individually z-score labels (within each session) before model fitting.

**Step 1**: Set the dimensionality of the unsupervised subspace to 2, and the dimensionality of the background subspace to 3 (an arbitrary choice that works well in our experience; this is another hyperparameter that can be adjusted).

**Step 2**: Set $\beta = 1$, $\delta = 50$, and fit models for $\alpha = [50, 100, 500, 1000, 5000]$. Choose the value of $\alpha$ that provides a desirable trade-off between frame reconstruction and label reconstruction (call this $\alpha'$).

**Step 3**: Set $\alpha = \alpha'$ and fit models for all combinations of $\beta = [1, 5, 10, 20]$ and $\delta = [10, 50, 100, 500]$. Choose the $\alpha$, $\beta$ combination with the lowest correlation coefficient averaged over all pairs of unsupervised dimensions (as in S13H Fig) (call these $\beta'$ and $\delta'$). The session classification accuracy may also be used to determine $\beta'$ and $\delta'$ (as in S13I Fig).

**Step 4** [optional]: Set $\alpha = \alpha'$, $\beta = \beta'$, $\delta = \delta'$ and refit the MSPS-VAE using several random weight initializations, which may result in qualitatively and/or quantitatively improved models (using latent traversals and correlation coefficients/session classification accuracies, respectively).

**Step 5** [optional]: Increase the dimensionality of the unsupervised subspace by 1 (leaving the dimensionality of the background subspace), then repeat Steps 2–4.

## Decoding labels from VAE latents

The VAE is a fully unsupervised method that does not take label information into account during training. After training, however, we can assess the degree to which the latent space of the VAE captures the label information by performing a post-hoc regression from the latent space to the labels. To do so we take the VAE latents and the user-supplied labels for all training trials and fit ridge regression models with a wide range of regularization values (0.01, 0.1, 1, 10, 100, 1000, 10000, 100000). We choose the best model using 5-fold cross validation, where each fold is constructed using just the training trials. We then evaluate the best model on each trial in the test data (e.g. Fig 2). We repeat this analysis using a multi-layer perceptron (MLP) neural network as a nonlinear regression model. The MLPs contain two hidden layers with 20 ReLU units each. Regularization, cross-validation, and evaluation are all performed as with the ridge regression models.

## Behavioral segmentation with autoregessive hidden Markov models

We fit two-state autoregressive hidden Markov models (ARHMMs) with the Expectation-Maximization (EM) algorithm using the `ssm` package [100]. We randomly initialize the discrete states, and then perform linear regression within each state to initialize model parameters. We train five models with different random initializations using 150 iterations of EM, and choose the model with the highest log-likelihood on the training data. The training data used to fit these models is the same set of training data used to fit the PS-VAE models.

## Decoding latents from neural activity

To decode the VAE and PS-VAE latents from neural activity we use an MLP neural network $f_{\mathrm{MLP}}$, which minimizes the mean square error (MSE) between predicted ($\hat{\mathbf{z}}_t$) and true ($\mathbf{z}_t$) latents (both supervised and unsupervised) at each time point $t$. The input to the decoder is a window of neural activity ($\mathbf{u}_t$) centered at time $t$ such that

$$\hat{\mathbf{z}}_t = f_{\mathrm{MLP}}(\mathbf{u}_{t-L:t+L}).$$

All hidden layers use ReLU nonlinearities, and contain the same number of units. We use stochastic gradient descent to train the models, using the Adam optimizer [98] with a learning rate of 1e-4. Training is automatically terminated when the running average of the MSE over the previous 10 epochs, computed on held-out validation data, begins to increase *or* training

**Table 4. Hyperparameter search details for decoding PS-VAE latents from neural activity.** Bolded entries indicate final values chosen through a hyperparameter search (S6 Fig). Some of the "Two-view by region" hyperparameters are region-specific, and not indicated here in the final row.

| | Hidden layers | Hidden unit number | Lags ($L$) | Learning rate |
|---|---|---|---|---|
| Mouse face | [1, **2**, 3, 4, 5] | [16, 32, **64**, 96, 128] | [1, 2, 4, 8, **16**] | **1e-4** |
| Mouse face ME | [1, 2, 3, **4**, 5] | [**16**, 32, 64, 96, 128] | 16 | **1e-4** |
| Two-view | [1, **2**, 3, 4, 5] | [16, 32, **64**, 96, 128] | 16 | **1e-4** |
| Two-view ME | [1, **2**, 3, 4, 5] | [16, 32, **64**, 96, 128] | 16 | **1e-4** |
| Two-view by region | [1, 2, 3] | [16, 32, 64] | 16 | [**1e-3**, 1e-4] |

reaches 200 epochs. The train/val/test data split used with these models is the same split used to fit the autoencoder models.

In addition to decoding the PS-VAE latents, we also decoded the motion energy (ME) of the latents (S6 Fig), as previous work has demonstrated that video ME can be an important predictor of neural activity [22, 23, 52].

We performed a hyperparameter search over the neural network architecture for each dataset and latent type (regular and ME), the details of which are shown in Table 4. We also decoded the true labels from neural activity (rather than the PS-VAE predictions of the labels; S8 Fig), as well as decoded the VAE latents from neural activity. For the label and VAE decoding we used the best hyperparameter combination from the corresponding PS-VAE latents in order to reduce the computational overhead of the hyperparamter search. We found in the mouse face dataset that increasing the number of lags $L$ continued to improve the model fits up to $L = 16$; therefore we chose to cap this hyperparameter due to our relatively small batch sizes ($T = 150$ to $T = 189$ time points). This finding is consistent with our previous work on the two-view dataset [27], so for all subsequent model fits we fixed $L = 16$, as reflected in Table 4. Dataset-specific decoding details are given below.

**Mouse face decoding**. To perform the decoding analysis on the mouse face data, we first took 10 random subsamples of 200 neurons (with replacement) from the original population of 1370 neurons. We performed this subsampling to reduce the high-dimensionality of the neural data, which allowed us to perform a larger, faster hyperparameter search. We then performed the hyperparameter search for each subsample. Next we computed the MSE on held-out validation data, and chose the set of hyperparameters that resulted in the best performing model on average across all subsamples (bolded in Table 4). Reconstructions in Fig 6 and S17 Fig use held-out test data that was neither used to train the model nor choose the best set of hyperparameters.

**Two-view decoding**. To perform the decoding analysis on the two-view data, we first used all 258 dimensions of neural activity returned by the LocaNMF algorithm [19]. We computed a bootstrapped version of the MSE on held-out validation data by randomly selecting 90% of the trials and computing the MSE, which we repeated (with replacement) 10 times. We then chose the set of hyperparameters that resulted in the best performing model on average across all bootstrapped samples (bolded in Table 4). Reconstructions in Fig 9 and S17 Fig use held-out test data that was neither used to train the model nor choose the best set of hyperparameters.

**Two-view region-based decoding**. We also decoded PS-VAE latents from region-specific neural activity, where the dimensions of neural activity ranged from 3 (TEa1 left/right hemispheres) to 24 (VIS left hemisphere) (see Table 1). We reduced the range of the hyperparameters to account for the reduced dimensionality of the input data, as well as to reduce computational overhead. We found that a larger learning rate (1e-3) was necessary for the

models to quickly converge. Results in Fig 9 use held-out test data that was neither used to train the model nor choose the best set of hyperparameters.

## Decoding behavioral videos from neural activity

To decode the behavioral videos themselves from neural activity (rather than just the latent representation) we proceed in two steps: first, we train an MLP neural network that maps from neural activity $\mathbf{u}$ to the PS-VAE latents $\mathbf{z}$; we denote the neural reconstructions as $\tilde{\mathbf{z}}$. Then, we train a convolutional decoder network $\tilde{g}(\cdot)$ that maps from the reconstructed latents $\tilde{\mathbf{z}}$ to video frames $\mathbf{x}$, producing reconstructed frames $\tilde{\mathbf{x}}$. This procedure improves upon the neural decoding performed in [27], which did not re-train the weights of the convolutional decoder; instead, the reconstructed latents were pushed through the frame decoder of the original VAE that produced the latents ($g(\cdot)$ in our notation; see Fig 1). However, the neural reconstructions of the latents contain noise not seen by $g(\cdot)$ during its training; retraining $g(\cdot)$ with the neural reconstructions to produce $\tilde{g}(\cdot)$ results in improved frame reconstructions.

In practice we fine-tune the weights of $g(\cdot)$ to get $\tilde{g}(\cdot)$. We construct a convolutional decoder neural network that has the same architecture as the PS-VAE (see Table 3 for an example) and initialize the weights with those of the PS-VAE frame decoder $g(\cdot)$. We then train the decoder for 200 epochs, using the PS-VAE latents predicted from neural activity on the training data. S25 and S26 Videos display video reconstructions from held-out test trials.

## Supporting information

**S1 Fig. The PS-VAE unsupervised latents and derived discrete states from an autoregressive hidden Markov model (ARHMM) agree with hand engineered feature detectors in the mouse face dataset. A**: *Top*: The red box outlines the frame crop used to compute a hand engineered whisker pad feature. We perform PCA on this cropped portion of each frame and take the first principal component (PC) as a proxy for whisker pad position. *Bottom*: The PS-VAE unsupervised latent corresponding to the whisker pad (Fig 4E) plotted against the whisker pad crop PC. **B**: Same as panel A, but using a different crop (in purple) to compute a hand engineered eyelid feature. The PS-VAE unsupervised "Eyelid" latent is highly correlated with this hand engineered feature. **C**: A 2-state ARHMM fit to the whisker pad PC produces a hand engineered whisker pad movement detector. **D**: The whisker pad movement detector constructed from the PS-VAE "Whisker pad" latent (reproduced from Fig 5). The continuous signals in panels C, D (and the discrete states derived from them, indicated by the background colors) are similar. **E**: A confusion matrix shows the overlap between the discrete states inferred from the PS-VAE "Whisker pad" latent and the whisker pad PC (each row adds to 1). **F**: The discrete states derived from the VAE latents are highly overlapping with those from the whisker pad movement detectors in panels C and D (reproduced from Fig 5). **G**: Overlap of states derived from the VAE latents and the whisker pad PC show the VAE-based states are highly correlated with whisker movements.
(PDF)

**S2 Fig. ARHMMs achieve better segmentation with fewer states and latents on simulated data. A**: *Top*: A 2D time series is generated using a 2-state ARHMM. The background color indicates the true discrete state at each time point. *Bottom*: A separate 2-state ARHMM is then fit to this simulated data. The inferred states visually match the true states well on this window of data. **B**: A confusion matrix shows the overlap between true and inferred states on held-out test data (each row adds to 1). The ARHMM is able to perfectly recover the discrete states. **C**: This process is repeated three more times to yield four independent time series. In each case

an ARHMM is able to perfectly recover the discrete states. **D**: *Top*: The four 2D time series from above are stacked to form an 8D time series. This results in data with $2^4 = 16$ discrete states (indicated by the background color), since each of the four independent time series can be in one of two states at each time point. *Bottom*: A 16-state ARHMM is then fit to this 8D simulated data, resulting in a mismatch between the true and inferred states on some time points. **E**: The confusion matrix shows many errors in the inferred states due to the larger dimensionality of the data and the larger number of states. By splitting the data into subsets of dimensions as in A and C and fitting a larger number of simple ARHMMs we recover the true discrete states more accurately.
(PDF)

**S3 Fig. Behavioral segmentations with 2-state ARHMMs based on low-dimensional PS-VAE outputs improve reliability and interpretability on the mouse face dataset, compared to ARHMMs fit with more dimensions or more states. A**: [Reproduced from Fig 5.] *Top*: Supervised PS-VAE latents corresponding to pupil location. Background colors indicate the states recovered by the 2-state saccade detector, which we call "pupil still" (light gray) and "pupil move" (dark gray). *Bottom*: Unsupervised PS-VAE latent corresponding to the whisker pad, and the states recovered by the 2-state whisking detector—"whisk still" (light gray) and "whisk move" (dark gray). **B**: The states from panel A combinatorially define four unique states (since each of the two ARHMMs can be in one of two states at each time point), which we refer to as the "combinatorial" PS-VAE states. **C**: The pupil location and whisker pad latents are concatenated and fit with a 4-state ARHMM. We refer to the resulting states as the "combined" PS-VAE states. There is general agreement between the combined and combinatorial states, although the combined states contain more state switches. **D**: A confusion matrix shows the overlap between between the combinatorial and combined states across all held-out test data. There remain many incongruous time points—for example, only 61% of the time points identified by the combinatorial state "pupil move/whisk move" is captured in a single combined state. **E**: A 4-state ARHMM is fit to the VAE latents. The resulting segmentation is somewhat aligned with the combinatorial PS-VAE segmentation in panel B but is much noisier, especially during whisker movements. **F**: There is poor overlap between the combinatorial PS-VAE states and the VAE states, suggesting that the VAE states are not capturing simple combinations of pupil and whisker movement. However, due to the lack of interpretability in the VAE latents, it is difficult to assess from this visualization alone what behaviors the VAE states capture.
(PDF)

**S4 Fig. The PS-VAE unsupervised latents and derived ARHMM states agree with hand engineered feature detectors in the two-view dataset.** Conventions and conclusions are the same as S1 Fig. **A**: The pink box outlines the frame crop used to compute a hand engineered jaw feature, which is highly correlated with the PS-VAE unsupervised latent corresponding to the jaw. **B**: The brown box outlines the crop used to compute a hand engineered chest feature, which is modestly correlated with the PS-VAE unsupervised latent corresponding to the chest. It is difficult to compute such a hand engineered feature that is not contaminated by the paw or the mechanical lever, demonstrating an advantage of the PS-VAE in this dataset. **C**: A 2-state ARHMM fit to the jaw and chest PCs (the "body" PCs) produces a hand engineered body movement detector. **D**: The body movement detector constructed from the PS-VAE body latents (reproduced from Fig 8). **E**: A confusion matrix shows the overlap between the discrete states inferred from the PS-VAE body latents and the body PCs. **F**: The discrete states derived from the VAE latents are highly overlapping with those from the body movement detectors in panels C and D (reproduced from Fig 8). **G**: Overlap of states derived from the

VAE latents and the body PCs suggests the VAE-based states are highly correlated with body movements.
(PDF)

**S5 Fig. Behavioral segmentations with 2-state ARHMMs based on low-dimensional PS-VAE outputs improve reliability and interpretability on the two-view dataset, compared to ARHMMs fit with more dimensions or more states.** Conventions and conclusions are the same as S3 Fig. **A**: [Reproduced from Fig 8.] *Top*: Supervised PS-VAE latents corresponding to paw location. *Bottom*: Unsupervised PS-VAE latents corresponding to the body. **B**: The states from panel A combinatorially define four unique states. **C**: The paw and body latents are concatenated and fit with a 4-state ARHMM. There is general agreement between the combined and combinatorial states, although the combined states contain more state switches. **D**: A confusion matrix shows the overlap between between the combinatorial and combined states across all held-out test data. There remain many incongruous time points— for example, only 63% of the time points identified by the combinatorial state "paw move/ body move" is captured in a single combined state. **E**: A 4-state ARHMM is fit to the VAE latents. The resulting segmentation is well aligned with the combinatorial PS-VAE segmentation in panel B, but tends to be noisier during body movements. **F**: There is poor overlap between the combinatorial PS-VAE states and the VAE states, suggesting that the VAE states are not capturing simple combinations of paw and body movements.
(PDF)

**S6 Fig. Results of hyperparameter searches for neural decoding models, which varied the number of hidden layers and number of hidden units per layer in an MLP neural network. A**: Hyperparameter search results for decoding PS-VAE latents from the mouse face data. The "best" model is indicated with an asterisk ($*$), and is a chosen to balance model performance and model complexity. Error bars represent a 95% bootstrapped confidence interval over 10 random subsamples of 200 neurons. **B**: $R^2$ results for the best model, separated by latent (reproduced from Fig 6C). Neural activity is able to successfully reconstruct the pupil area, eyelid, and horizontal position of the pupil location. The poor reconstruction of the vertical position of the pupil location may be due to the small dynamic range (and accompanying noise) of that label. The boxplot represents variability in $R^2$ over the 10 random subsamples. **C**: Hyperparameter search results for decoding PS-VAE latent motion energy (ME) from the mouse face data. Error bars as in A. **D**: $R^2$ results for the best model, separated by latent. The ME of the whisker pad is decoded well, consistent with the results in [22] and [23]. Boxplot variability as in B. **E-H**: Same as A-D, except on the two-view dataset. The mechanical equipment (Lever, L Spout, R Spout), which has low trial-to-trial variability (Fig 8A), is decoded better than the animal-related latents. The accuracy of the ME decoding is similar across all latents. Error bars represent a 95% bootstrapped confidence interval over test trials; boxplots represent variability across 10 bootstrapped samples from the test trials (see Methods for more information).
(PNG)

**S7 Fig. Quantification of neural decoding performance using all available brain regions for the two-view dataset. A**: Decoding accuracy ($R^2$) computed separately for each PS-VAE latent demonstrates how the PS-VAE can be utilized to investigate the neural representation of different behavioral features (same as S6F Fig). **B**: Decoding accuracy computed separately for each VAE latent. Boxplots show variability over 10 bootstrapped samples from the test trials (see Methods for more information).
(PDF)

**S8 Fig. Comparison of neural decoding performance for the true labels and their corresponding supervised latents.** Decoding accuracy for the true labels ($x$-axis) and their corresponding supervised latents in the PS-VAE ($y$-axis), for both the mouse face dataset (panel **A**) and the two-view dataset (panel **B**); individual dots represent the median over test trials using different subsamples of neurons (panel A) or trials (panel B). The decoding accuracy is very similar across most labels in both datasets, indicating that the noise introduced in the PS-VAE label reconstruction does not have a large effect on the neural decoding of these quantities. The model architecture used for decoding the true labels is the same as the best model architecture found for decoding the PS-VAE latents (S6 Fig).
(PNG)

**S9 Fig. The supervised subspace of the MSPS-VAE successfully reconstructs the labels across all training sessions. A**: Frames from each of the four sessions used to train the models. **B**: The true labels (black lines) are almost perfectly reconstructed by the supervised subspace of the MSPS-VAE (blue lines). We also reconstruct the labels from the latent representation of a single VAE trained on all sessions (orange lines), which captures some features of the labels but misses much of the variability. **C**: Observations from the individual batches in panel B hold across all labels and test trials for each sesson. Error bars represent a 95% bootstrapped confidence interval over test trials.
(PDF)

**S10 Fig. PS-VAE hyperparameter selection for the freely moving mouse dataset.** Panel descriptions are the same as those in Fig 11.
(PDF)

**S11 Fig. PS-VAE hyperparameter selection for the mouse face dataset.** Panel descriptions are the same as those in Fig 11.
(PDF)

**S12 Fig. PS-VAE hyperparameter selection for the two-view dataset.** Panel descriptions are the same as those in Fig 11.
(PDF)

**S13 Fig. MSPS-VAE hyperparameter selection for the multi-session head-fixed mouse dataset. A**: MSE per pixel as a function of latent dimensionality and the hyperparameter $\alpha$, which controls the strength of the label reconstruction term. The frame reconstruction is robust across many orders of magnitude. **B**: MSE per label as a function of latent dimensionality and $\alpha$. Subsequent panels detail $\beta$ and $\delta$ with an 11D model ($|\mathbf{z}_s| = 4$, $|\mathbf{z}_u| = 4$, $|\mathbf{z}_b| = 3$) and $\alpha$ = 50. **C**: MSE per pixel as a function of $\beta$ and $\delta$; frame reconstruction is robust to both of these hyperprameters. **D**: MSE per label as a function of $\beta$ and $\delta$; label reconstruction is robust to both of these hyperprameters. **E**: Index code mutual information as a function of $\beta$ and $\delta$. **F**: Total Correlation as a function of $\beta$ and $\delta$. **G**: Dimension-wise KL as a function of $\beta$ and $\delta$. **H**: Average of all pairwise Pearson correlation coefficients in the model's 4D unsupervised subspace as a function of $\beta$ and $\delta$. **I**: Session classification accuracy from a linear classifier that predicts session identity from the unsupervised latents, as a function of $\beta$ and $\delta$. **J**: Triplet loss as a function of $\beta$ and $\delta$. Error bars in panels A-D represent 95% bootstrapped confidence interval over test trials; line plots in panels E-J are the mean values over test trials, and confidence intervals are omitted for clarity.
(PDF)

**S14 Fig. Frame reconstruction video still. First (top) row**: frames from the original behavioral video. **Second row**: reconstructed frames from the PS-VAE (*left*), residuals between these

reconstructed frames and the original frames (*center*), and the corresponding PS-VAE latents (*right*). Colors are consistent with those in the main figures, and latent names are provided in the legend above. **Third row**: reconstructed frames from a standard VAE, the residuals, and the corresponding VAE latents. **Fourth row**: frames reconstructed from the labels only, without unsupervised latent dimensions (again with the residuals and the corresponding true labels). Reconstructions are shown for several batches of test data not used for training or model selection. Each batch is separated by black frames in the reconstructions and breaks in the traces.
(PNG)

**S15 Fig. Latent traversal video still.** Each panel contains latent traversals for a single dimension, the name of which is indicated in the lower left corner of the panel. Supervised latents with associated labels use the label names; unsupervised latents that we have applied a post-hoc semantic label to are indicated with quotations, such as "Whisker pad" and "Eyelid" here. Unsupervised latents that have not been given labels (for example the VAE latents) are named Latent 0, Latent 1, etc. (none of this type are shown here). The supervised dimensions that correspond to 2D spatial locations contain an additional red dot that signifies the desired position of the body part, seen here in the Pupil (x) and Pupil (y) panels. A missing dot indicates a low-likelihood label that was omitted from the objective function. The latent traversal procedure uses a base frame, indicated in the upper left corner of the figure. The videos show traversals for a range of base frames, and traversals for different base frames are separated by several black frames.
(PNG)

**S16 Fig. ARHMM movement detector video still. Top**: examples of video are played from 5 frames before to 5 frames after the transition from the still state to the moving state, as identified by the ARHMM. The state for each frame is indicated by the text in the bottom left corner. Different examples are separated by several black frames. **Bottom**: Corresponding traces of the PS-VAE latents used to fit the ARHMM; the latents corresponding to the current example are displayed in bold.
(PNG)

**S17 Fig. Neural decoding video still. Top left**: frames from the original behavioral video. **Top center**: reconstructed frames from the PS-VAE. **Top right**: frames reconstructed by the neural activity. **Bottom left**: residual between the PS-VAE and neural reconstructions. **Bottom right**: PS-VAE latents (gray traces) and their predictions from neural activity (colored traces; colors are consistent with those in the main figures). Reconstructions are shown for several batches of test data, which were not used for the training or model selection of either the PS-VAE or the neural decoders. Each test batch is separated by black frames in the frame reconstructions, and breaks in the traces.
(PNG)

**S18 Fig. MSPS-VAE session swap video still.** On the left are MSPS-VAE reconstructions from the original behavioral video. Each remaining panel shows frames reconstructed from the same set of latents, except the background latents have been set to the median value of the background latents for the indicated session.
(PNG)

**S1 Video. Frame reconstruction video for head-fixed mouse dataset.** Refer to S14 Fig for captions.
(MP4)

**S2 Video. Frame reconstruction video for freely moving mouse dataset.** Refer to S14 Fig for captions.
(MP4)

**S3 Video. Frame reconstruction video for mouse face dataset.** Refer to S14 Fig for captions.
(MP4)

**S4 Video. Frame reconstruction video for two-view dataset.** Refer to S14 Fig for captions.
(MP4)

**S5 Video. Frame reconstruction video for multi-session head-fixed mouse dataset.** Refer to S14 Fig for captions.
(MP4)

**S6 Video. PS-VAE latent traversal video for head-fixed mouse dataset with $\beta = 5$ ("good" disentangling).** 2D unsupervised subspace. Refer to S15 Fig for captions.
(MP4)

**S7 Video. PS-VAE latent traversal video for head-fixed mouse dataset with $\beta = 1$ ("bad" disentangling).** 2D unsupervised subspace. Refer to S15 Fig for captions.
(MP4)

**S8 Video. PS-VAE latent traversal video for head-fixed mouse dataset.** 3D unsupervised subspace. Refer to S15 Fig for captions.
(MP4)

**S9 Video. PS-VAE latent traversal video for head-fixed mouse dataset.** 4D unsupervised subspace. Refer to S15 Fig for captions.
(MP4)

**S10 Video. VAE latent traversal video for head-fixed mouse dataset.** Refer to S15 Fig for captions.
(MP4)

**S11 Video. $\beta$-TC-VAE latent traversal video for head-fixed mouse dataset.** Refer to S15 Fig for captions.
(MP4)

**S12 Video. PS-VAE latent traversal video for freely moving mouse dataset.** Refer to S15 Fig for captions.
(MP4)

**S13 Video. VAE latent traversal video for freely moving mouse dataset.** Refer to S15 Fig for captions.
(MP4)

**S14 Video. PS-VAE latent traversal video for mouse face dataset.** Refer to S15 Fig for captions.
(MP4)

**S15 Video. VAE latent traversal video for mouse face dataset.** Refer to S15 Fig for captions.
(MP4)

**S16 Video. PS-VAE latent traversal video for two-view dataset.** Refer to S15 Fig for captions.
(MP4)

**S17 Video. PS-VAE latent traversal video for two-view dataset.** Only the mechanical equipment was tracked in this dataset (leaving the single visible paw untracked). We increased the dimensionality of the unsupervised latent space and show that these latents now capture the untracked paw. Refer to S15 Fig for captions.
(MP4)

**S18 Video. VAE latent traversal video for two-view dataset.** Refer to S15 Fig for captions.
(MP4)

**S19 Video. PS-VAE latent traversal video for multi-session head-fixed mouse dataset.** Refer to S15 Fig for captions.
(MP4)

**S20 Video. MSPS-VAE latent traversal video for multi-session head-fixed mouse dataset.** Refer to S15 Fig for captions.
(MP4)

**S21 Video. ARHMM saccade detector for mouse face data.** Refer to S16 Fig for captions.
(GIF)

**S22 Video. ARHMM whisker pad movement detector for mouse face data.** Refer to S16 Fig for captions.
(GIF)

**S23 Video. ARHMM paw movement detector for two-view data.** Refer to S16 Fig for captions.
(GIF)

**S24 Video. ARHMM body movement detector for two-view data.** Refer to S16 Fig for captions.
(GIF)

**S25 Video. Neural decoding video for mouse face dataset.** Refer to S17 Fig for captions.
(MP4)

**S26 Video. Neural decoding video for two-view dataset.** Refer to S17 Fig for captions.
(MP4)

**S27 Video. MSPS-VAE session swap video for multi-session head-fixed mouse dataset.** Refer to S18 Fig for captions.
(MP4)

**S1 Appendix. Related work.**
(PDF)

**S1 Table. Hyperparameter details for models presented in the main text figures.** $|\mathbf{z}_s|$, $|\mathbf{z}_u|$, and $|\mathbf{z}_b|$ denote the dimensionality of the supervised, unsupervised, and background latent spaces. *We used the orthogonalization procedure outlined in the MSPS-VAE Methods section to remove $\gamma$ as a hyperparameter for this model.
(PDF)

## Acknowledgments

We thank Anne Churchland for helpful comments on the manuscript. We also thank the following for making their data publicly available: Matteo Carandini and Ken Harris (mouse

face), and Simon Musall and Anne Churchland (two-view mouse). Finally, we thank Olivier Winter, Julia Huntenburg, and Mayo Faulkner for helpful comments on the code.

## Author Contributions

**Conceptualization:** Matthew R. Whiteway, Dan Biderman, Liam Paninski.

**Data curation:** Matthew R. Whiteway, Yoni Friedman, E. Kelly Buchanan, Anqi Wu, John Zhou, Niccolò Bonacchi, Michael Schartner.

**Formal analysis:** Matthew R. Whiteway, Michael Schartner.

**Funding acquisition:** Mario Dipoppa, C. Daniel Salzman, John P. Cunningham, Liam Paninski.

**Investigation:** Mario Dipoppa, Nathaniel J. Miska, Jean-Paul Noel, Erica Rodriguez, Karolina Socha, Anne E. Urai.

**Methodology:** Matthew R. Whiteway, Dan Biderman, E. Kelly Buchanan, Anqi Wu.

**Resources:** John Zhou.

**Software:** Matthew R. Whiteway, Dan Biderman, Yoni Friedman, E. Kelly Buchanan, Anqi Wu, John Zhou.

**Writing – original draft:** Matthew R. Whiteway, Dan Biderman, Liam Paninski.

**Writing – review & editing:** Matthew R. Whiteway, Dan Biderman, Mario Dipoppa, E. Kelly Buchanan, Anqi Wu, Jean-Paul Noel, John P. Cunningham, Liam Paninski.

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
