## [Decision Letter · Decision Letter 0]

16 Jun 2021

Dear Dr Whiteway,

Thank you very much for submitting your manuscript "Partitioning variability in animal behavioral videos using semi-supervised variational autoencoders" for consideration at PLOS Computational Biology. As with all papers reviewed by the journal, your manuscript was reviewed by members of the editorial board and by several independent reviewers. The reviewers appreciated the attention to an important topic. Based on the reviews, we are likely to accept this manuscript for publication, providing that you modify the manuscript according to the review recommendations.

Dear Authors,

As you will see, we were fortunate to get 4 reviewers to look at your submission. Most of the comments are easy to address. There might be two exceptions which are the generalizability to non-head fixed behaviors and the qualitative assessment of the "ease of interpretation" (see Reviewer3). As mentioned by reviewer 4 you should consider revising your title regarding generalizability or demonstrating that indeed this approach is useful in freely behaving animals.

Best wishes,

Frederic

Sincerely,

Frédéric E. Theunissen

Associate Editor

PLOS Computational Biology

Thomas Serre

Deputy Editor

PLOS Computational Biology

[LINK]

Dear Authors,

As you will see, we were fortunate to get 4 reviewers to look at your submission. Most of the comments are easy to address. There might be two exceptions which are the generalizability to non-head fixed behaviors and the qualitative assessment of the "ease of interpretation" (see Reviewer3). As mentioned by reviewer 4 you should consider revising your title regarding generalizability or demonstrating that indeed this approach is useful in freely behaving animals.

Best wishes,

Frederic

Reviewer's Responses to Questions

**Comments to the Authors:**

Reviewer #1: The authors present an extension of the VAE which makes use of hand-labeled data in order to produce untangled latent representations that may aid in behavioral quantification. The proposed PS-VAE forces a subset of the latent representations to contain label information and encourage the additional latent dimensions to capture features of behavior that are independent from the labeled inputs.

Overall this work is a very interesting take on feature ‘untangling’ in low-dimensional latent representations of behavior, and is of interest from both a behavior interpretability a well as deep learning perspective. Movies demonstrating these partitions by generating frames along a single one of the unsupervised dimensions indicate that this method can successfully discover behavioral features from movies that would not be possible using output of posture estimation alone.

The idea of using the supervised and unsupervised subspaces is elegant and explained well in the manuscript and the details of training are sufficient for reproduction. Additionally, the authors have generated and submitted code that appears well-documented.

Here I list a few weaknesses or simply aspects of the manuscript that might be presented more clearly:

Introduction:

The idea of disentangling latent dimensions in image data is not new and prior work might deserve more of a mention in the introduction (Zheng, Zhilin, and Li Sun. "Disentangling latent space for vae by label relevant/irrelevant dimensions." Proceedings of the IEEE/CVF Conference on Computer Vision and Pattern Recognition. 2019, for example).

Results:

The authors mention that traditional methods are limited to constrained tasks, but then only go on to explore videos in head-fixed animals. Demonstrating the viability of this method in a less constrained environment that captures more of the body and a higher diversity of behavior - where simple tracking within the image would not suffice to describe pose - would greatly strengthen these claims.

2.2

The comparison of the vanilla VAE results to the true x and y coordinates does not seem like a fair one here and in section 2.3.1. Is it possible to compare the emergent behavior detectors that come from the unsupervised latent space with specifically engineered detectors?

2.3.2

Authors mention ‘meaningful features’ in the context that small pixel changes be of greater interest than large pixel variance, but do not address the sampling that that could capture rare events of interest. Is this considered in any accuracy metric? For example a rare event that is known to be important to animal but only occurs for a single frame?

The head-fixed example of capturing interpretable unsupervised representations is an exciting addition and should be emphasized - what happens when there are fewer or more unsupervised latent dimensions? How did the authors reach the decision to use only two additional dims? Is it trivial to assign meaning to any number of these latents?

When separating x-y limb motion or mechanical equipment motion, does the method fail when using slightly offset imaging angles?

Discussion:

How is ‘salience of features’ addressed, are there examples where the unsupervised space did not produce immediately interpretable features but may contain information that may be useful to quantify?

Again, how relevant is this for phenotyping where individuals may have physical discrepancies in addition to behavioral differences?

The authors mention freely moving behavior, but how difficult would it be to apply these methods to recordings with many views where the animal takes up only a small portion of the visual field at any time? Would translating information into an animal-centric coordinate system be feasible before training the PS-VAE? How sensitive would this preprocessing have to be?

Methods:

Is the data used for each example only coming from one individual and one camera configuration? If so are there ways to make this a viable approach for datasets which are recorded across days and individuals and might have drift or lighting differences? If not then this can be made more clear.

Specific comments:

Line 49 - consist instead of consists?

Reviewer #2: Whiteway et al introduce Partitioned Subspace VAEs, a modification of the classical VAE architecture that they show can extract interpretable features of animal movement from tracked videos. PS-VAEs learn a latent representation that separates out video signals arising from tracked features (here, positions of anatomically defined keypoints) from other sources of variance, for example movement of the jaw or chest. They then demonstrate how these identified latents can be further analyzed using AR-HMMs to identify behavioral events, and how latents can be related to neural activity. The authors apply the PS-VAE to three example datasets, in each case contrasting their model with a standard VAE to show the gains in interpretability their model achieves. The authors develop several helpful metrics to demonstrate this contrast, such as the variance of PS-VAE latents aligned to behavioral events. Another strength of this paper is its clear organization and writing, including the Methods section, which does a very nice job of walking the reader through the details of the PS-VAE.

Key points to address:

First, several other papers have proposed semi-supervised versions of VAEs, including modifications to enhance the interpretability of learned latent representations. The authors note the contrast between PS-VAEs and these other methods in the paper Discussion, however I believe the paper needs a more detailed discussion of the differences between the PS-VAE and some of these other models, perhaps by the addition of a Related Works section in the Introduction.

Ideally, it would also be great to also see the PS-VAE compared to some of these other VAE variations (like the pi-VAE) in the Results section, to show what the specific design of the PS-VAE has achieved beyond what was already possible via other methods. Are the results shown here really something that wouldn’t be possible with other, related methods? Or is this just the first study to try this kind of analysis on videos from behavioral neuroscience? Alternatively, the authors might include an ablation study to show how the different components of the loss function collectively contribute to the performance of the PS-VAE.

An obvious question that arose in reading this paper was how the results depend on the dimensionality of the unsupervised latent space. This is a user-provided parameter, and in the paper with one exception it’s always set to two (the one exception being a case where two supervised latents were removed, and the number of unsupervised latents was increased from two to four to compensate.) How should an end-user select the dimensionality of the unsupervised latent space? Is there a way to tell that you’ve done a good job? If you over- or under-estimate the dimensionality of the unsupervised latents, does interpretability of the results suffer?

In the Discussion, the authors describe several possible extensions of the PS-VAE, including its application to freely behaving animals and to neural data. As a more general extension: does this model require the supervised latents to come from tracked positions of keypoints? Or could these inputs take a more general form? Similarly, must the input to the PS-VAE be video data, or could other signals also be used? I understand that the target audience of this paper is neuroscientists using tracking methods on behavior videos, however the PS-VAE seems general enough that it could be relevant to other types of data. While it would be great to see this (or some of the other proposed extensions) addressed with a brief example in the results section, I understand if the authors consider this to be out of scope. Instead, it might be helpful for the authors to include some kind of general summary/figure on the types of data to which the PS-VAE might be applied.

Finally, I had one more specific comment, motivated by the two-view mouse dataset. Here the authors state that “the PS-VAE provides a substantial advantage over the VAE for any experimental setup that involves moving mechanical equipment.” While I agree that the PS-VAE seems likely to outperform the VAE in most settings, it seems that in order for the PS-VAE to successfully isolate equipment-derived signals, it must be possible to predict the appearance of equipment from one or more tracked keypoints. I would guess that there are some types of equipment (deformable objects like mouse bedding? the black/white patterned balls used for fly-on-a-ball experiments?) where keypoint-based prediction of equipment appearance could fail. I suggest the authors add a section to the Discussion on what kinds of signal can vs cannot be learned by the supervised latents.

Reviewer #3: Review is uploaded as an attachment.

Reviewer #4: Dear Editors,

The manuscript under review, entitled “Partitioning variability in animal behavioral videos using semi-supervised variational autoencoders” by Whitway et al, provides a new computational method termed Partitioned Subspace Variational Autoencoder (PS-VAE). PS-VAE is a variation of semi-supervised modelling based on a fully unsupervised variational autoencoder, which provides the benefit of both a supervised and unsupervised component - allowing for a more full description of both supervised pose-estimation and unsupervised analysis of variability that is not captured by pose-estimation.

This is a strong approach for the analysis of pose-estimation based behavioral analysis, which has certainly gained a great deal of popularity with the release of pose-estimation architectures like SLEAP and DLC. However, the applicability of this method to behavior other than highly-controlled head-fixed rodent setups remain to be seen.

However, within head-fixed setups, the authors do a great job of validating their approach. The authors use their method in 3 datasets, including one from the IBL, to address the applicability of their approach across different head-fixed behavioral setups. Their approach, although perhaps only applicable pragmatically to head-fixed mice, will certainly be of interest to labs using head-fixed behavioral preparations.

I am impressed with the approach the authors present in this paper. The overall approach is clear, the documentation is acceptable, and inclusion of examples of varying pose-estimation architectures is welcomed.

I have only one major comments:

Based on the current title, I was expecting a more generalizable approach. I strongly suggest the title be changed to better reflect what is shown in the paper. “Partitioning variability in head-fixed rodent behavioral videos using semi-supervised variational autoencoders” would be a much better title, until the inclusion of data suggesting that this method can be applied to freely moving mouse behaviors.

**Have the authors made all data and (if applicable) computational code underlying the findings in their manuscript fully available?**

Reviewer #1: Yes

Reviewer #2: Yes

Reviewer #3: Yes

Reviewer #4: Yes

PLOS authors have the option to publish the peer review history of their article (what does this mean?). If published, this will include your full peer review and any attached files.

Reviewer #1: No

Reviewer #2: No

Reviewer #3: No

Reviewer #4: No

Figure Files:

Data Requirements:

Reproducibility:

References:

---

## [Editor Report · Decision Letter 1]

9 Sep 2021

Dear Dr Whiteway,

We are pleased to inform you that your manuscript 'Partitioning variability in animal behavioral videos using semi-supervised variational autoencoders' has been provisionally accepted for publication in PLOS Computational Biology.

Best regards,

Frédéric E. Theunissen

Associate Editor

PLOS Computational Biology

Thomas Serre

Deputy Editor

PLOS Computational Biology

Dear authors,

Thank you for carefully addressing all of the concerns of the reviewers. Congrats on a nice contribution in quantitative analyses of behavior.

Frederic Theunissen

---

## [Editor Report · Acceptance letter]

15 Sep 2021

PCOMPBIOL-D-21-00850R1 

Partitioning variability in animal behavioral videos using semi-supervised variational autoencoders

Dear Dr Whiteway,

I am pleased to inform you that your manuscript has been formally accepted for publication in PLOS Computational Biology. Your manuscript is now with our production department and you will be notified of the publication date in due course.

With kind regards,

Olena Szabo
